

# Wind Transport of Snow Impacts Ka- and Ku-band Radar Signatures on Arctic Sea Ice

Vishnu Nandan[1,2*], Rosemary Willatt[3], Robbie Mallett[3], Julienne Stroeve[1,3], Torsten Geldsetzer[2], Randall Scharien[4], Rasmus Tonboe[5], Jack Landy[6], David Clemens-Sewall[7], Arttu Jutila[8], David N. Wagner[9,10], Daniela Krampe[8], Marcus Huntemann[11], John Yackel[2], Mallik Mahmud[2], David Jensen[1], Thomas Newman[3], Stefan Hendricks[8], Gunnar Spreen[11], Amy Macfarlane[9], Martin Schneebeli[9], James Mead[12], Robert Ricker[13], Michael Gallagher[14], Claude Duguay[15,16], Ian Raphael[7], Chris Polashenski[7], Michel Tsamados[3], Ilkka Matero[8], and Mario Hoppmann[8]

[1]Centre for Earth Observation Science (CEOS), University of Manitoba, Canada
[2]Department of Geography, University of Calgary, Canada
[3]Centre for Polar Observation and Modeling, University College London, UK
[4]Department of Geography, University of Victoria, Canada
[5]DTU Space, Technical University of Denmark, Denmark
[6]Centre for Integrated Remote Sensing and Forecasting for Arctic Operations (CIRFA), UiT The Arctic University of Norway, Tromsø, Norway
[7]Thayer School of Engineering, Dartmouth College, USA
[8]Alfred Wegener Institute Helmholtz Centre for Polar and Marine Research, Bremerhaven, Germany
[9]WSL Institute for Snow and Avalanche Research (SLF), Davos, Switzerland
[10]CRYOS, School of Architecture, Civil and Environmental Engineering, EPFL, Lausanne, Switzerland
[11]Institute of Environmental Physics, University of Bremen, Germany
[12]ProSensing Inc, Amherst, MA, USA
[13]Norce Norwegian Research Centre AS, Bergen, Norway
[14]Physical Sciences Laboratory, NOAA, USA
[15]Department of Geography and Environmental Management, University of Waterloo, Canada
[16]H2O Geomatics Inc., Waterloo, Canada

*Correspondence*: Vishnu Nandan (Vishnu.Nandan@umanitoba.ca)



**Abstract:** Wind transport alters the snow topography and microstructure on sea ice through snow redistribution controlled by deposition and erosion. The impact of these processes on radar signatures is poorly understood. Here, we examine the effects of snow redistribution on Arctic sea ice from Ka- and Ku-band radar signatures. Measurements were obtained during two wind events in November 2019 during the MOSAiC expedition. During both events, changes in Ka- and Ku-band radar waveforms and backscatter coincident with surface height changes measured from a terrestrial laser scanner are observed. At both frequencies, snow redistribution events increased the dominance of the air/snow interface at nadir as the dominant radar scattering surface, due to wind densifying the snow surface and uppermost layers. The radar waveform data also detect the presence of previous air/snow interfaces, buried beneath newly deposited snow. The additional scattering from previous air/snow interfaces could therefore affect the range retrieved from Ka- and Ku-band satellite radar altimeters. The relative scattering contribution of the air/snow interface decreases, and the snow/sea ice interface increases with increasing incidence angles. Relative to pre-wind conditions, azimuthally averaged backscatter at nadir during the wind events increases by up to 8 dB (Ka-band) and 5 dB (Ku-band). Binned backscatter within 5° azimuth bins reveals substantial backscatter variability in the radar footprint at all incidence angles and polarizations. The sensitivity of the co-polarized phase difference is linked to changes in snow settling and temperature-gradient induced grain metamorphism, demonstrating the potential of the radar to discriminate between newly deposited and older snow on sea ice. Our results reveal the importance of wind, through its geophysical impact on Ka- and Ku-band radar signatures of snow on sea ice and has implications for reliable interpretation of airborne and satellite radar measurements of snow-covered sea ice.



## 1 Introduction

Wind plays an important role in shaping the spatial distribution of snow depth and snow water equivalent (SWE) over sea ice (Moon et al., 2019; Iacozza & Barber, 2010). Wind alters snow temperature gradients (Colbeck, 1989), structural anisotropy (Leinss et al., 2020), and snow grain growth (Löwe et al., 2007). Furthermore, wind affects the residence and sintering time

of snow close to the surface, facilitating depositional snow dune growth and erosional processes (Trujillo et al., 2016). Fluctuating wind speeds and directions thus modify snow surface topography and density via wind scouring and compaction of snow (Lacroix et al., 2009). Depending on the ice surface roughness (e.g., level ice, pressure ridges, hummocks etc.), wind will result in the formation of heterogeneously distributed cm-scale ripple marks to snow bedforms and drifts on the scale of 10's of meters (Filhol & Sturm, 2015; Sturm et al., 1998). This further alters the geometric, aerodynamic, and radar-scale

roughness on sea ice (Savelyev et al., 2006; Fung & Eom, 1982).

Since wind redistribution of snow impacts snow depth distribution and SWE, this can in turn alter Ka- and Ku-band radar backscatter signatures used in the airborne- and satellite-based retrievals of sea ice freeboard and thickness. Under cold and calm snow conditions, a common assumption in radar altimetry is that the Ka-band co-polarized radar signal returns originate from the air/snow interface (e.g., Lawrence et al., 2018), and the Ku-band returns originate from the snow/sea ice interface

(e.g., Tilling et al., 2018), due to dominant surface scattering from these interfaces (Fung & Eom, 1982). For synthetic aperture radar (SAR) and scatterometry, variations in snow grain microstructure or from inclusions within the sea ice influence the proportion of surface and volume scattering to the total radar backscatter (Fung, 1994). Winds can roughen/smoothen the snow surface, inducing additional Ka- and Ku-band surface and/or volume scattering contributions to the dominant scattering surfaces and radar backscatter.

Very little is known about how wind redistributed snow impact snow depth, SWE, and thickness retrievals from airborne and satellite radars (e.g., Yackel & Barber, 2007; Kwok & Cunningham, 2008; Kurtz et al., 2009; Kurtz & Farrell, 2011; Glissenaar et al., 2021). Due to repeat airborne and satellite ground-tracks often occurring weeks/months apart and the drift motion of sea ice, it is challenging to measure radar backscatter changes between calm and wind-affected snow cover, both on the same parcel of sea ice over time. For example, Kurtz & Farrell (2011) assumed snow redistribution events as the cause

for the anomalous snow depth decrease in 2009 over multi-year sea ice in the Canadian Archipelago (CA), retrieved from two Operation IceBridge (OIB) snow radar flights, acquired three weeks apart. Yackel & Barber (2007) speculated that snow redistribution events on first-year sea ice in the CA caused a change in SWE up to 7 cm, derived from two C-band RADARSAT-1 images, acquired 45 days apart. To represent small-scale spatial variability due to snow redistribution that are not captured in the large-scale satellite products, studies have developed snow redistribution functions, designed for high-

resolution laser altimetry data (e.g., Kwok & Cunningham, 2008; Kurtz et al., 2009). However, Glissenaar et al. (2021) applied the snow redistribution scheme developed by Kurtz et al. (2009) on the OIB-derived radar freeboard and snow depth products for the Arctic Ocean and found no correlation between radar freeboard and snow depth estimates averaged over the



footing-scale of the CryoSat-2 (300 m) and ENVISAT (2 km) radar altimeters. They concluded that applying a snow redistribution scheme on radar altimetry freeboard data would not improve the sea ice freeboard-to-thickness conversion.

Overall, to better understand the impact of snow redistribution on Ka- and Ku-band radar signatures, we require unambiguous *in-situ* measurements of snow physical properties and meteorological observations during wind events, sampled coincident with surface-based radar measurements. This bridges a fundamental knowledge gap towards improved modelling of Ka- and Ku-band radar waveforms and backscatter at multiple polarizations and incidence angles, as well as to better interpret Ka- and Ku-band radar signatures from presently operational SARAL/AltiKa (Guerreiro et al., 2016), CryoSat-2 (Lawrence et

al., 2018), Sentinel-3 (Lawrence et al., 2021), ScatSat-1 (Singh & Singh, 2020) and the upcoming Ka-/Ku-band CRISTAL altimetry (Kern et al., 2020) and SWOT satellite missions (Armitage & Kwok, 2021).

In our study, we investigate wind-induced changes in snow physical properties and topography on Ka- and Ku-band dominant scattering surfaces and backscatter using a surface-based, fully-polarimetric, Ka- and Ku-band radar (KuKa radar; see Stroeve et al., 2020) that was deployed during the 2019-20 Multidisciplinary drifting Observatory for the Study of Arctic Climate

(MOSAiC) expedition (Krumpen et al., 2020). Here, we present the analysis of data gathered between 9 and 16 November 2019, assessing the effects of two separate Wind Events ('WE1' and 'WE2'). First, we describe the KuKa radar system, the time series of meteorological observations, snow physical properties, and snow surface topography. Next, we investigate the impact of snow redistribution on Ka- and Ku-band radar echograms and waveforms, examining changes in dominant scattering surfaces, radar backscatter and co-polarized phase difference. Finally, we discuss our findings, relevant towards

improving retrievals of snow/sea ice geophysical variables from airborne and satellite radars.

## 2. Data and Methods

### 2.1 Surface-Based Ka- and Ku-band Polarimetric Radar (KuKa Radar)

During the MOSAiC expedition, the German research icebreaker *R/V Polarstern* drifted with a sea ice floe across the central Arctic Ocean over a full annual cycle (See Figure 1 in Nicolaus et al., 2022). The floe was dominated by second-year ice including ~ 60% refrozen melt ponds (Krumpen et al., 2020). The Remote Sensing Site (RSS) was first established on the

floe on 18 October 2019, where the KuKa radar was deployed on ~ 80 cm thick, homogenous, and undeformed sea ice.





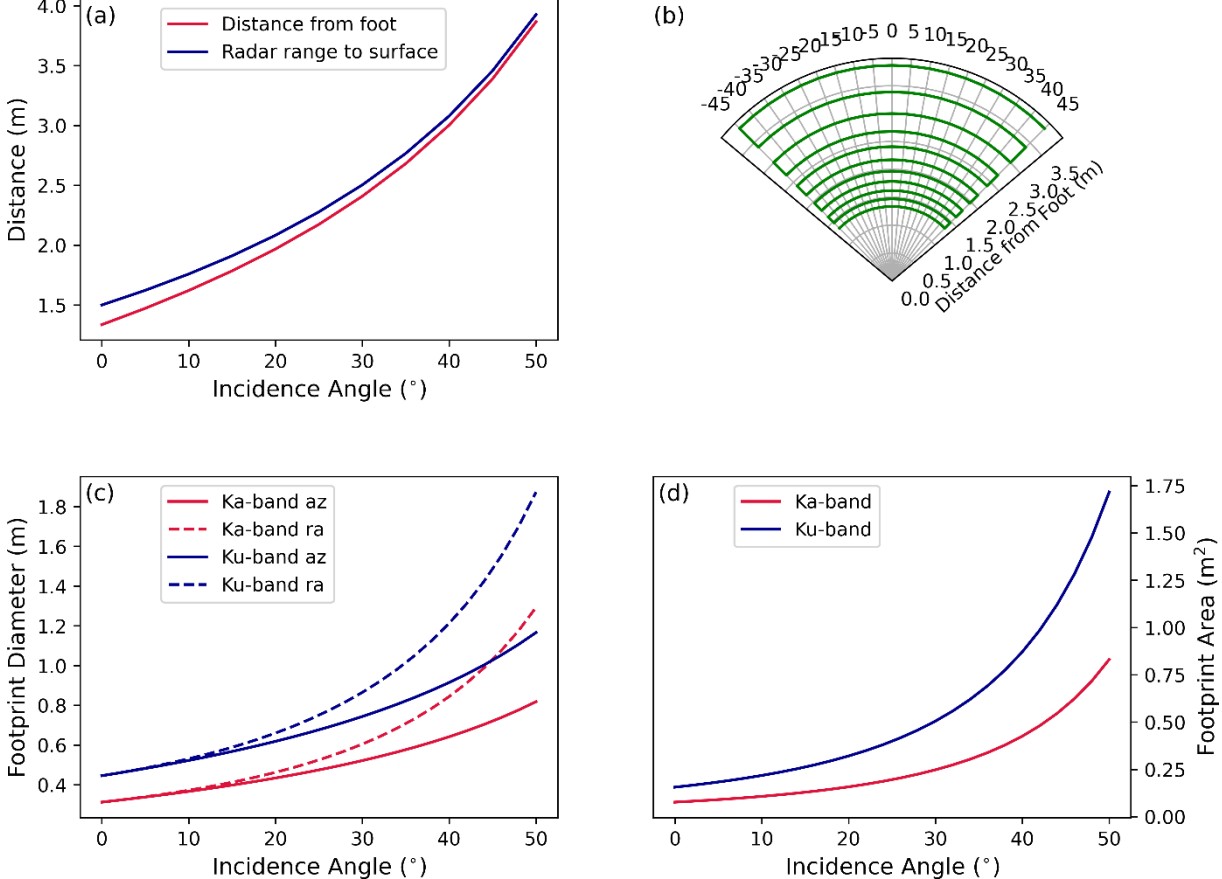

Figure 1: KuKa radar geometry illustrating (a) radial distance and radar range from the pedestal; (b) scan pattern of radar
projected onto a level surface; (c) diameter of radar footprint, measured radially ('ra') and azimuthally ('az'); and (d) area of
radar footprint.

The KuKa radar transmits at Ka- (30-40 GHz) and Ku-band (12-18 GHz) frequencies and measures the normalized radar cross
section per unit area (NRCS) or total radar backscatter, expressed in decibel (dB) (Stroeve et al. 2020). The radar acquires
several independent samples across a fixed azimuth ($\theta_{az}$) range, at discrete incidence angle ($\theta_{inc}$) intervals. The radar measures
all vertical (V) and horizontal (H) linear polarization transmit and receive combinations: VV, HH, HV, and VH. As such, it is
fully-polarimetric, enabling derivation of many polarimetric parameters including the co-polarized phase difference (CPD),
analysed here.

The centre frequency was set to match the Ka-band of AltiKa (35 GHz) and the Ku-band of CryoSat-2 (13.575 GHz). The
KuKa radar bandwidth is considerably higher than the bandwidth of AltiKa and CryoSat-2, allowing improved range resolution



of 1.5 cm for Ka-band and 2.5 cm for Ku-band relative to 30 cm and 46 cm for AltiKa and CryoSat-2, respectively. The radial

distance and range from the pedestal, the footprint diameter, and footprint area from nadir to $\theta_{inc} = 50°$ are shown in Figure

1. The radar is beam-limited and, given the 11.9° and 16.9° antenna beamwidths at Ka- and Ku- bands, respectively, the size

of the radar footprint on the snow is dependent on frequency, height of the antenna above the snow surface, and $\theta_{inc}$. This also

means that the Ka- and Ku-band footprint overlap for a given radar 'shot' is $\theta_{inc}$ dependent. The overlapping footprint is

between -5° to +45° $\theta_{az}$ for Ku-band, and -45° to +5° $\theta_{az}$ for Ka-band. Further description of the radar specifications, signal

processing, polarimetric calibration routine, signal-to-noise and error estimation is documented in Stroeve et al. (2020).

At the RSS, the radar acquired scans every 30 mins over a 90° $\theta_{az}$ width, called a scan line, between nadir and $\theta_{inc} = 50°$ at

5° increments. Between 9 and 15 November, a total of 325 scans were collected. Following WE2, the ice supporting the RSS

broke up on 16 November, and the measurements were stopped until it was safe to redeploy the radar.

**2.2 Meteorological and Snow Property Data**

A 10-m tall meteorological station installed ~ 100 m away from the RSS monitored 2 m air and surface temperature (°C),

relative humidity (%), air pressure (hPa), wind speed (m/s) and wind direction (°). Wind direction is denoted with respect to

geographic north (0°). Measurements were acquired and logged every second (Cox et al., 2021) and resampled to 30-minute

averages, to match the radar scan intervals.

A thermal infrared (TIR) camera (Infratec VarioCam HDx head 625, assuming emissivity 0.97 at 7.5-14 μm wavelength)

(Spreen et al., 2022) measured snow surface temperature (°C), every 10 minutes. Two digital thermistor chains (DTC) installed

close to the RSS measured near-surface, snow, and sea ice temperature evolution at 2 cm vertical intervals. No destructive

snow sampling was done underneath the KuKa radar footprint. Instead, snow depth measurements were sampled using a metre

stick close to the radar on 4 and 14 November. Profiles of the penetration resistance force of the snow were collected before,

during and after WE1 and WE2 using a SnowMicroPen (SMP; Johnson & Schneebeli, 1999) at the 'Snow1', 'Snow2' and the

RSS sites (see locations in Figure 4). Five SMP profiles per pit were recorded weekly. To compare initial density and SSA

between the RSS and the Snow1 and Snow2 locations at the beginning of November, one SMP profile from the RSS was taken

on 4 November. The force profiles were converted into density and specific surface area (SSA) following King et al. (2020)

and Proksch et al. (2015) parameterizations, respectively, that worked well for snow during the MOSAiC winter (Wagner et

al., 2021).

**2.3 Snow Surface Topography**

Footage from a visual surveillance closed-circuit television (CCTV) camera was used to visualize snow surface topography

changes within the radar footprint (Spreen et al., 2021). In addition, Terrestrial Laser Scanning (TLS; Deems et al., 2013) data

were collected on 1, 8 and 15 November using a Riegl VZ1000, which provided point clouds of the snow surface topography

in the radar footprint. Scan positions were registered in RiSCAN (Riegl's data processing software) using reflectors



permanently frozen to the ice and levelled based on the VZ1000's built-in inclination sensor. Wind-blown snow particles were removed from the data by FlakeOut filtering (Clemens-Sewall et al, in review). Filtered data were aligned to one another by matching reflectors and other tie-points. To transform the TLS data into the KuKa radar's reference frame, the outlines of the radar's pedestal column and the antenna arms were manually picked in the TLS data.


A non-linear least squares optimization method using SciPy (Virtannen et al 2020) was then implemented to estimate the best fitting circle and rectangle to match the pedestal column and the antenna arms, respectively. The centre of the pedestal was used as the horizontal origin, the centre of the antennas was used for orientation, and the antenna height at nadir position was used as the vertical origin. Within the radar's reference frame, a polar grid was defined with radial increments of 0.25 m and azimuthal increments of 10°. The surface height in the radar reference frame (a.k.a. the vertical distance from the surface to the radar antennas at nadir) for each grid cell was calculated by averaging the vertical position of each TLS point within that grid cell.

### 2.4 Radar Waveform, Backscatter and Co-polarized Phase Difference

Radar waveform analysis is performed to determine how WE1 and WE2 affected the surfaces and volumes detected by the radar, especially the dominant scattering surface. Waveforms from each sampling time across the $\theta_{az}$ range are recorded and overlaid with the TLS data to aid in interpretation of where in the snow/sea ice the Ka- and Ku-band backscatter originated from (Section 3.2). For the waveform analysis, deconvolved waveforms are used (described in Stroeve et al., 2022). To summarise, data are deconvolved using waveforms from a refrozen lead located close to the RSS in January 2020 (see Stroeve et al, 2020), which provided a specular return useful for reducing the appearance of sidelobes that result from non-ideal behaviour of the RF electronics, as well as internal reflections in the radar. Waveforms are stacked horizontally to form a echogram, demonstrating how the return waveforms changed over WE1 and WE2 from the overlapping footprint. The NRCS from the range power profiles are calculated following the standard beam-limited radar range equation (Ulaby et al., 2014), given by:

$$NRCS = \frac{8 ln\ (2)h^2 \sigma_c}{\pi R_C^4 \theta_{3dB}^2 cos\ \theta} \left( \frac{\widetilde{P_r}}{\widetilde{P_{rc}}} \right)$$

where $h$ is the antenna height, $R_C$ is the range to the corner reflector, $\theta_{3dB}$ is the one-way half-power beamwidth of the antenna and $\widetilde{P_r}$ and $\widetilde{P_{rc}}$ are the received power from the snow and the corner reflector, respectively.

The peak power in the radar waveforms used for calculating NRCS is determined by locating the highest peak in the waveform averaged across all polarisations. The NRCS value is calculated based on the power contained within this peak, by integrating over the range bins where the power falls below a threshold, are set to -50 dB on either side of the peak for Ka-band data, and -20 dB (-40 dB) on the on the smaller-range (larger-range) sides for Ku-band data. The return power is integrated over the entire snow volume, so the NRCS values include scattering contributions at the air/snow and snow/ice interfaces as well as





from within the snow volume. First, we calculate the NRCS values for the air/snow and snow/ice interfaces by integrating the power over the waveform peaks within +/- 2 dB either side from the overlapping footprint area (Section 3.2). Next, we calculate the NRCS averaged across the entire 90° $\theta_{az}$ range, at discrete $\theta_{inc} = 0°$, 15°, 35° and 50°, to demonstrate footprint-scale variability in backscatter during the two wind events (section 3.3).

To investigate the sub-footprint scale backscatter variability caused by surface heterogeneity, as well as the range to the dominant scattering surface that could have changed before, during and after WE1 and WE2, we use azimuth 'sectoring' and analyse the NRCS averaged at 5° wide $\theta_{az}$ bins (i.e., negative $\theta_{az}$ sectors between -45° to -40°… -5° to 0° and positive $\theta_{az}$ sectors between 0° to +5°… +40° to +45°) (Figure 1b). Azimuth 'sectoring' has an impact on the number of independent samples along a 5° $\theta_{az}$ bin, since a smaller area is used for averaging (Table 1). The number of independent samples is estimated by dividing half the antenna beamwidth by the $\theta_{az}$ width, then multiplied by the number of range gates (Geldsetzer et al., 2007).

**Table 1**: Number of independent samples at Ka- and Ku-band frequencies at nadir and $\theta_{inc} = 50°$ at $\theta_{az} = 90°$ and 5°

| Frequency | Nadir | | $\theta_{inc} = 50°$ | |
|---|---|---|---|---|
| | $\theta_{az} = 90°$ | $\theta_{az} = 5°$ | $\theta_{az} = 90°$ | $\theta_{az} = 5°$ |
| Ka-band | 487 | 48 | 1609 | 439 |
| Ku-band | 198 | 34 | 1252 | 376 |

Within every $\theta_{inc}$ scan, VV, HH and HV are derived from the complex covariance matrix (second-order derivative of the scattering matrix containing amplitude and phase), while VH is discarded based on the observed reciprocity of cross-polarized channels (i.e., HV ~ VH) (Ulaby et al., 2014). We also use the derived co-polarized phase difference (CPD) given by arctan $\left[\frac{Im\langle S_{HH}S_{VV}^*\rangle}{Re\langle S_{HH}S_{VV}^*\rangle}\right]$ uniformly distributed over $[-\pi, \pi]$ (Ulaby et al., 2014). CPD is sensitive to the snow structural anisotropy changes (e.g. Leinss et al., 2016) resulting from snow residence and settling time, as well as snow metamorphic change resulting from the snow temperature gradient (Löwe et al., 2011; Leinss et al., 2020). Studies on terrestrial snow from X-band SAR show that new snow has a horizontal alignment of snow crystals that results in greater anisotropy and CPD (i.e., positive phase shift) (Voglimacci-Stephanopoli et al., 2022; Leinss et al., 2016;). With subsequent temperature-gradient induced metamorphism, the growth of vertical structures overpowers the build-up of horizontal structures during snow settling, decreasing the anisotropy and CPD (i.e., negative phase shift) (Leinss et al., 2016). In section 3.3.2, we show the changes in backscatter signatures and CPD variability across the KuKa radar footprint at 5° wide $\theta_{az}$ bins at specific timestamps on 9, 11 and 15 November.



## 3. Results

### 3.1 Meteorological and Snow Conditions

#### 3.1.1 WE1 and WE2

The floe experienced two wind events between 11 and 16 November 2019. WE1 occurred on 11 November and lasted until ~ 0800 UTC on 12 November when winds ~12 m/s blew from SW to SE directions (Figure 2). WE2 started ~ 0900 UTC on 15 November, when a low-pressure system began to intensify (Figure 3a). The wind direction shifted from SW to W, and speeds increased to ~ 15 m/s and continued until ~ 1900 UTC on 16 November (Figure 2). During WE2, the low-pressure system dropped just below 995 hPa (Figure 3a) and air temperatures reached as high as -5.5°C. The warm air advection was

accompanied by a steep increase in relative humidity that reached > 90% (Figure 3b).



Figure 2: Line plots illustrate daily, 30-min averages of 2 m air temperature (MET tower) and snow surface temperature measurements from the TIR camera, MET tower and DTC sensors; acquired between 9 and 16 November. Wind rose plots illustrate corresponding wind speed (m/s) and direction (°) measurements recorded by the MET tower. All times are UTC.





### 230    3.1.2 Snow Temperature, Density and Microstructure

Figure 3: Line plots show daily, 10-min averaged 2 m (a) air pressure and (b) relative humidity, recorded by the MET tower between 9 and 16 November. Surface plots show DTC-derived hourly-averaged temperature gradient of (c) near-surface, snow, sea ice and ocean; and (d) snow volume from the RSS. Yellow pixels represent snow volume. DTC temperature sensors are spaced every 2 cm, with the top 20 cm

representing the distance between the first sensor located above the air/snow interface and at the air/snow interface. Red and orange boxes indicate WE1 and WE2 window. Note different temperature gradient scales for (c) and (d)








Figure 4: The upper 10 cm of the horizontally averaged density and SSA profiles of the snowpack over time derived from the SMP force
signals (where the average consists of 5 SMP profiles at each location), from (a & b) Snow 1 - A1, (c & d) Snow 1 - A5, and (e & f) Snow
2 - A2 locations. In each subplot, the horizontally averaged profile measured at the RSS measured on 4 November 2019 is shown for
comparison (blue dashed line). Map shows the immediate surroundings of the study site. The RSS is shown with a red dot, colored lines
show the extent of Snow1 and Snow2 sites, and SMP locations within these sites in colored shapes. The background is preliminary
quicklook-processed surface elevation data from the airborne laser scanner, where the whiter colors indicate high elevations of $\geq 2$ m.

During WE1, the snow surface temperature increased on 11 November from ~ -32°C (0800 UTC) to ~ -16°C (~ 2000 UTC)
(Figure 2). During WE2, snow surface temperature increased to ~ -4°C by ~ 1800 UTC on 15 November and remained
relatively warm until the end of WE2 (Figure 2). A large temperature gradient of ~ 3°C/cm was observed during WE1, whereas
the gradient decreased by half during WE2 (Figure 3c & d).  During this period, snow temperature gradients consistently
exceeded 0.25°C/m, suggesting temperature gradient-driven hoar metamorphism was occurring throughout the snowpack (e.g.
Colbeck, 1989).

SMP-derived density and SSA profiles measured at all Snow1 and Snow2 locations exhibit an increase and decrease in density
and SSA over time, respectively, from the uppermost snow layers (Figure 4). An increase in snow density in the uppermost 2
cm layer in the snowpack is visible for all three locations (left panels). The increase at Snow 1 - A5 until 26 November is most
distinct. The density and SSA profile from the RSS measured on 4 November correlates well with those from Snow1 and
Snow2, indicating representative snowpack evolution conditions between RSS and Snow1 and 2 locations. The average density
change of the upper 2 cm between the last and the first measurement at each location is +30.7 kg/m$^3$ at Snow 1 - A1, +79.3
kg/m$^3$ at Snow 1 - A5, and +22.9 kg/m$^3$ at Snow 2 - A2 (Figure 4). The SSA change is -2.0 mm$^{-1}$ at Snow 1 locations, and -
2.0 mm$^{-1}$ at Snow 2 - A2 (right panels). Based on the 5 SMP profiles, we computed snow depth changes, where we found a
slight increase for each location. At Snow 1, the increase was 1.7 cm and 0.2 cm at A1 and A5 locations, respectively, with a
1.2 cm increase in snow depth from the A5 location, sampled between 4 and 26 November. At Snow2 - A2, the overall increase
was 0.3 cm, with a 0.8 cm increase recorded between 13 and 20 November.

The increase (decrease) in snow depths indicate snow deposition (erosion) processes. The increase in snow surface density is
typical for strong wind action on the snow (Lacroix et al., 2009; Savelyev et al., 2006). Substantially warmer air temperatures
during the observed wind events, compared to pre-wind conditions (Figure 2) also increase the likelihood for snow grains to
sinter (e.g., Colbeck, 1989), favouring snow surface compaction. An SSA decrease indicates the reduction in surface area,
caused by the breakup of snow particles during wind transport (King et al., 2020).



### 3.1.3 Snow Surface Topography Dynamics

**3.1.3.1 Snow bedform evolution**

WE1 and WE2 resulted in a dynamic evolution of snow bedform features in the radar footprint (Figure 5 and Supplemental Video 1). On 9 and 10 November (Figure 5a & b), the snow cover is characterized by bedding features (white stars) in negative $\theta_{az}$ sectors, as well as crag and tail features and patterned tail markings in positive $\theta_{az}$ sectors (yellow star). The major axis of these bedforms is predominantly oriented parallel to the radar azimuthal scan direction. These features are typically found

on relatively level sea ice (Filhol & Sturm, 2015).

Between 11 November until ~ 0800 UTC on 12 November, winds blew snow both radially and azimuthally to the radar footprint. Because the radar sled forms an aerodynamic obstacle, the snow drifted unevenly in the lee of the sled (red star in Figure 5c-f and Supplemental Video 1). While snow depth was not measured directly below the KuKa radar, considering the 30 cm radar sled height, snow drifts covering the edges of the sled indicate an increase in snow depth to > 30 cm directly in

front of the radar. Blowing snow buried the existing bedforms from 9 and 10 November, creating a new drift, with its major axis oriented parallel to the positive $\theta_{az}$ sectors, and with an increasing slope (greater snow depth) with increasing $\theta_{inc}$ (black star in Figure 5e-g). A new sastrugi also developed in the negative $\theta_{az}$ sectors (brown star in Figure 5e & f). WE2 on 15 November caused the rapid formation of two new snow drifts in the negative $\theta_{az}$ sector, oriented parallel to the prevailing wind direction (purple stars in Figure 5g). A small pit-like feature also formed in the depression between the two drifts (dark

blue star in Figure 5g), while the drift (black star) that formed during WE1 is still visible in the positive $\theta_{az}$ sectors.







Figure 5: CCTV images from the RSS footprint between (a) 9 November and (g) 15 November. CCTV images were selected during times of the day when the ship's floodlight was illuminating the footprint. The KuKa radar is on the far right on the images, while an L-band Scatterometer is on the upper right. Coloured stars represent major snow bedforms within the KuKa radar footprint, while orange arrows show the orientation of the bedforms in response to prevailing wind direction. All times are UTC.

**3.1.3.2 Snow Surface Heights from TLS**

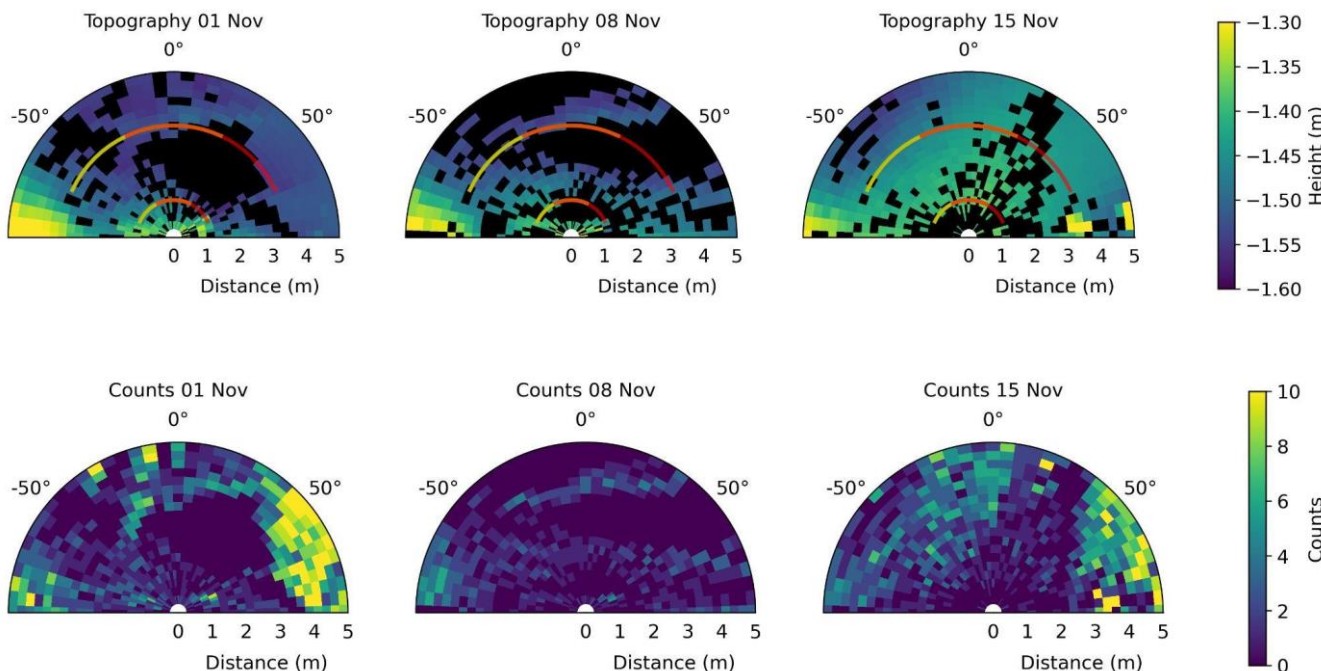

Figure 6: TLS data (plan view) from 1, 8 and 15 November, from -90° to + 90°, where the angle indicates the azimuth of the radar positioner, and radial horizontal distance measured from the centre of the radar pedestal. The top panels show the topography as measured downwards (increasing negative) from the middle of the radar antenna arms. Black indicates no data recordings in that bin. Projections of the centres of the radar footprints are shown for 0° and 50° radar inclination angles, superimposed on the TLS data in yellow and red for radar observations, respectively, and orange where the two overlap. The bottom panels indicate the number of TLS data points within each bin. Surface depressions resulting in 0 counts in the TLS data are due to obscuration by adjacent high areas due to snow/sea ice topography and human-made objects, as viewed from the TLS's oblique viewpoint some distance away.



The TLS-derived snow surface height data from 1, 8 and 15 November are shown in Figure 6 along with superimposed red, orange, and yellow lines, indicating the centres of the radar footprint. Data from 1 November are included for context (left panel), indicating that the surface topography was similar to 8 November (middle panel). The TLS data show considerable surface height variability within the radar footprint between 8 and 15 November, with snow surface height increasing (middle and right panel), as also indicated by the raised snow drift (black star in Figure 5e-g) at approximately 0° to 45° azimuth in the CCTV images.

## 3.2 Radar Waveforms

Figure 7 shows the progression of Ka- and Ku-band radar waveforms at nadir, overlaid with spatially coincident TLS-derived surface heights. The TLS data and the waveforms are both averaged into individual 5° azimuth sectors, with the highest peak power overlaid in blue. In the supplement, we provide an animation (Supplemental Video 2) that includes all radar data obtained during the two wind events. In this section, we only show and discuss four date/time frames to illustrate the radar response.

Prior to WE1, radar waveforms from 9 and 10 November (top left and right panels in Figure 7) remained stable, with only small power variations in each azimuthal bin over time. The air/snow interface detected by the radar corresponds to the heights detected by the TLS on 1 and 8 November, indicating that both Ka- and Ku-band frequencies detect the air/snow interface as the dominant scattering surface at VV and HH in most $\theta_{az}$ bins.

A lower scattering interface is also visible at ~ 20 to 40 cm below the air/snow interface, especially prominent in the HV data in both frequencies, but also visible in the VV and HH data. To understand this, we consider the HV waveform characteristics and local snow depth. Snow depth measured behind the KuKa radar footprint during 4 and 14 November varied between 21 and 29 cm (not shown). Note that these measurements were taken close to the instruments to not disturb the radar measurements, and therefore, snow depth in the radar footprint may differ (see also Figure 6 for snow height variability). The range values indicated in the radar waveforms are based on the speed of light in free space, and the speed of propagation of the EM radiation would reduce to approximately 80% of that value in the snow (Willatt et al., 2009). Taking this correction into account and assuming similar snow depths at nadir, the lower interface in the waveforms lay ~ 16 to 32 cm below the air/snow interface. Considering that snow depths were not directly measured in the footprint this is a good agreement. Based on the very small amount of radiation scattered from larger ranges, considering little penetration of Ku- and Ka-band signals into sea ice (Fung et al., 1994), and the consistency with local snow depth, we can conclude that this interface in the HV data is the snow/ice interface. A small amount of radiation is expected from ranges beyond this interface caused by snow and ice backscattering from the perimeter of the 30-50 cm radar footprint and sidelobes.









Figure 7: Progression of Ka- and Ku-band radar waveforms at nadir. Range (y-axis) is given from the antenna phase centre, and the antenna azimuth angles (x-axis) are the angles for that individual antenna. Plots showing VV, HH and HV are stacked vertically for the Ku- and Ka-band data in each panel set. Panels correspond to different dates/times on 9, 10, 11 and 15 November. The highest power peak (averaged across all polarisations) is indicated with a blue line, and the surface height in the spatially coincident TLS data is superimposed on top (coloured circles).

During WE1, radar waveforms at nadir in Figures 7 and 8 show that the peak power at the air/snow interface shifted upwards due to snow deposition at ~ 1800 UTC on 11 November (Figure 5c). This is followed by a snow scouring/erosion event, which is seen in the downward movement of the peak power (Supplemental Video 2), and then a second deposition event at approximately 0800 UTC on 12 November (Figure 5d), which again sees an upward movement of the peak power (Figure 8). It is interesting to note that the Ka- and Ku-band scattering can still be seen from the previous air/snow interface from 9 and 10 November, as well as from the snow/ice interface, more prominent in the Ku-band. After WE1, the new air/snow interface remains the dominant scattering surface for all polarizations and $\theta_{az}$ sectors.

During WE2, after accumulation of newly redistributed snow, the air/snow interface moved upwards to a closer range from the antenna phase centre (bottom right panel in Figures 7 and 8). Scattering from the previously detected air/snow interface (corresponding to the TLS data from 1 and 8 November) is still visible in both Ka- and Ku-band data (Figure 8). In addition, the air/snow interface from 11 November remains visible in the Ka-band data in all polarisations (bottom left panel in Figure 7).

Next, we examine the highest amplitude peak (under which the backscatter is calculated) at nadir, and how this varies with frequency and polarisation, through time. Prior to WE1, depending on the $\theta_{az}$ sector, the highest power peak fluctuated between the air/snow and snow/ice interfaces at both frequencies (top panels in Figure 7), suggesting variability in snow density (Figure 4) and surface topography (Figure 5) across the $\theta_{az}$ sector within the nadir footprint. During and after WE1 and WE2, the highest peak power remains almost always at the air/snow interface for both frequencies (bottom panels in Figure 7). This means that the backscatter values in the following Figures 8 to 10 correspond to the air/snow or snow/ice interfaces, depending on the $\theta_{az}$ sector and $\theta_{inc}$; i.e., changes in backscatter could correspond to scattering from different interfaces, rather than a change in backscatter from one interface.







Figure 8: Progression of the KuKa radar waveforms over the commonly sampled area -5° to +45° $\theta_{az}$ for Ku-band, and -45° to +5° $\theta_{az}$ for Ka-band). The top panels indicate the full time series from 2-15 November with the current air/snow, buried previous air/snow, and snow/ice interfaces indicated in red, yellow, and black, respectively. Sketched yellow arrows show how buried air/snow interfaces remain visible through time. Individual air/snow and snow/sea ice interface NRCS values are determined by integrating power between the red/black dashed/dotted lines, which cover the range bins where the power is within 2 dB from the air/snow and snow/sea ice interface peak. Time series of the interface NRCS values are shown below the echograms. The bottom panels show a temporal 'zoom in' of WE1. Right panels show line plots of the waveforms at the given times corresponding to the vertical dashed lines on the echogram.

Figure 8 demonstrates the effect of WE1 and WE2 on HH-polarized waveform shapes at nadir using echoes averaged across the Ka- and Ku-band overlapping area. HH data show that the air/snow interface is always the dominant scattering surface in both frequencies. In the HV data, the snow/ice interface is the dominant scattering surface, but both interfaces are visible in both frequencies and all polarisations. Previous air/snow interfaces are also visible as in Figure 7. The sketched yellow arrows on the Ku-band HH plot show how the previous air/snow interfaces remain visible when additional snow accumulates on top and remain visible throughout the timeseries. These buried interfaces, along with the snow/ice interface, appear at greater range when covered with thicker snow due to the reduced wave propagation speed in snow relative to air, increasing the two-way travel time back to the radar receiver.

For the Ka- and Ku-band HH data, there are relatively small changes to the NRCS associated with the snow/ice interface. However, changes to the NRCS associated with the air/snow interface are much larger; prior to WE1, the Ka-band air/snow interface NRCS reduces from -5 to -10 dB before increasing during and following WE1 to -3 dB. At Ku-band a similar pattern is observed with the air/snow NRCS reducing from -5 to -8 dB, then increasing to -3 dB following WE1. This indicates that most of the observed changes to overall NRCS during and after WE1 and WE2 relate to backscatter changes from the air/snow interface and only minimally to the snow/ice interface. The Ka-band HV data show the air/snow interface NRCS decreasing prior to WE1, increasing during the wind events and then reducing to a lower value than previously, whilst the Ku-band data show the air/snow interface NRCS increasing during the wind events and remaining higher than previously. The different behaviour at the two frequencies indicates that this could relate to roughness, i.e., the change in roughness is dependent on length scales. This is shown in further detail in the waveform line plots which indicate how the waveform shape changed with more variability relating to the air/snow interface and snow above the snow/ice interface in both frequencies and polarisations. Both the Ka- and Ku-band HV show the snow/ice interface becoming brighter during the wind events and remaining brighter afterwards, however, we are not able to confirm whether temperature gradient-driven snow metamorphism caused this.



### 3.3 Radar Backscatter and Co-Polarized Phase Difference

The waveform analysis described in Section 3.2 illustrates how the locations of the peak power evolved during WE1 and WE2. We now focus on the backscatter response by analyzing the azimuthally-averaged Ka- and Ku-band backscatter time series

over the overlap area, at discrete $\theta_{inc}$ = 0°, 15°, 35° and 50°. Included in the analyses are radar echograms at $\theta_{inc}$ = 15° and 35° during WE1 over the overlap area, to support backscatter interpretation at higher $\theta_{inc}$. Next, we make 2D interpolations of the spatial radar response along $\theta_{inc}$ and across $\theta_{az}$ at 5° $\theta_{az}$ bins over both Ka- and Ku-band footprints separately and analyse backscatter changes and CPD variability at specific times on 9, 11 and 15 November.

### 3.3.1 Azimuthally-averaged Backscatter

During pre-wind conditions, both Ka- and Ku-band backscatter are relatively stable at all $\theta_{inc}$ (Figure 9a & b). VV and HH backscatter primarily originates as surface scattering at the air/snow interface, and secondarily from snow volume scattering and surface scattering at the snow/sea ice interface. HV backscatter originates primarily from the snow/sea ice interface (top panels in Figure 7).

During WE1, nadir backscatter increases significantly, with a greater Ka-band increase of ~ 8 dB (VV and HH), compared to a Ku-band increase of ~ 5 dB (VV and HH) (Figure 9a & b). The waveform analysis in Figures 7 and 8 indicates that the amount of scattering from the snow/sea ice interface changed very little during WE1, while the scattering contribution to the backscatter from the air/snow interface increased significantly due snow redistribution, modifying the snow density (Figure 4) and surface/interface roughness (Figure 5). This increase is accompanied by additional VV and HH backscatter from the

previous, now-buried air/snow interface from the pre-wind conditions (Figure 8). HV peak power shifts from the snow/sea ice interface to the air/snow interface and the buried within-snow interface (Figure 8). This is clearly seen in the two significant HV increases at nadir, by up to 5 dB (Ka-band) and by up to 4 dB (Ku-band) during WE1 (Figure 9a & b), coinciding with two short-term snow depositional events at ~ 1800 UTC on 11 November and around 0700 UTC on 12 November (Figure 5c & d and Supplemental Video 1).


Figure 9: Azimuthally averaged (a) Ka- and (b) Ku-band backscatter at 0°, 15°, 35° and 50° incidence angles between 9 and 16 November. Red and orange indicate the WE1 and WE2 time window. Yellow circles correspond to times of the day (in UTC) when the CCTV camera captured snapshots of radar scans. Panels (c) and (d) show time series of Ka- and Ku-band radar echograms at (c) $\theta_{inc} = 15°$ and (d) $\theta_{inc} = 35°$ during WE1.





At $\theta_{inc}$ = 15° and 35°, the peak power interfaces during WE1 are much less obvious than at nadir but do exist (Figure 9c & d). However, the bulk of the peak power moves from the air/snow interface to the snow/sea ice interface at all polarizations. The shifting of peak power to the snow/sea ice interface coincides with a decrease in Ka-band VV and HH backscatter by up to 2 dB at $\theta_{inc}$ = 15°. The effect is less at $\theta_{inc}$ = 35° due to the reduced effect of air/snow interface roughness. The waveform

analysis shows that the relative contribution of the snow/sea ice interface becomes more important at shallow angles and the air/snow interface becomes relatively less prominent. This feature is more observable in the HV data where the air/snow interface scattering is subtle, and the snow/sea ice interface is brighter, with potential volume scattering from the snow grains (middle panels in Figure 9c & d). Ku-band at non-nadir incidence angles show negligible change in backscatter (more stable in HV at $\theta_{inc}$ = 35° and 50°), compared to Ka-band and pre-wind conditions (Figure 9b).


During WE2, Ka- and Ku-band backscatter at all $\theta_{inc}$ remains relatively stable (Figure 9a & b). Around ~ 2100 UTC on 15 November, a short-term snow depositional event (Supplemental Video 1) causes the Ka-band nadir backscatter to increase by ~ 2 dB. The Ka-band waveform analysis shows scattering contributions from the air/snow interface during the snow deposition and from previously detected air/snow interface from 11 November (Figure 8 and lower right panels in Figure 7), causing the

additional 2 dB increase. Similar to WE1, Ku-band backscatter at $\theta_{inc}$ = 35° and 50° almost remains the same throughout WE2 (Figure 9b). Next, we show changes in the spatial varying backscatter and co-polarized phase difference signatures within each 5° $\theta_{az}$ sector acquired at specific date/times during pre-wind conditions, WE1 and WE2.


### 3.3.2 Backscatter Response and Co-Polarized Phase Difference at $\Delta\theta_{az} = 5°$

#### 3.3.2.1 Change in Backscatter

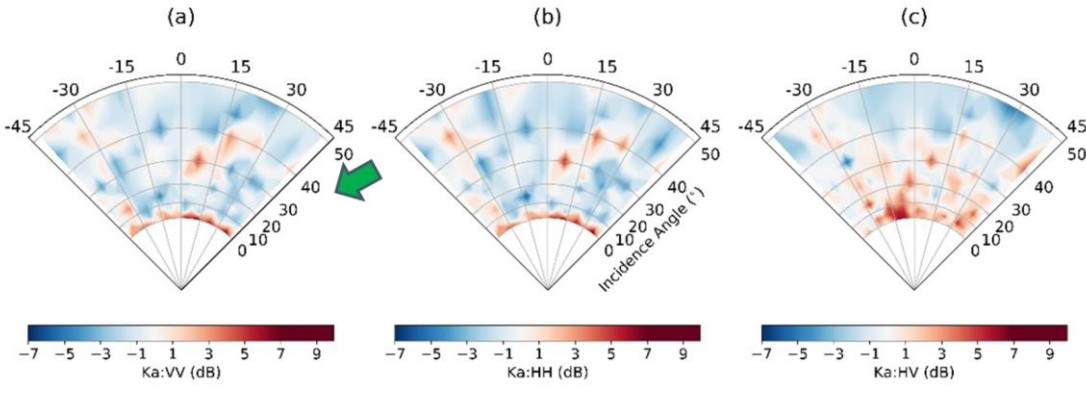

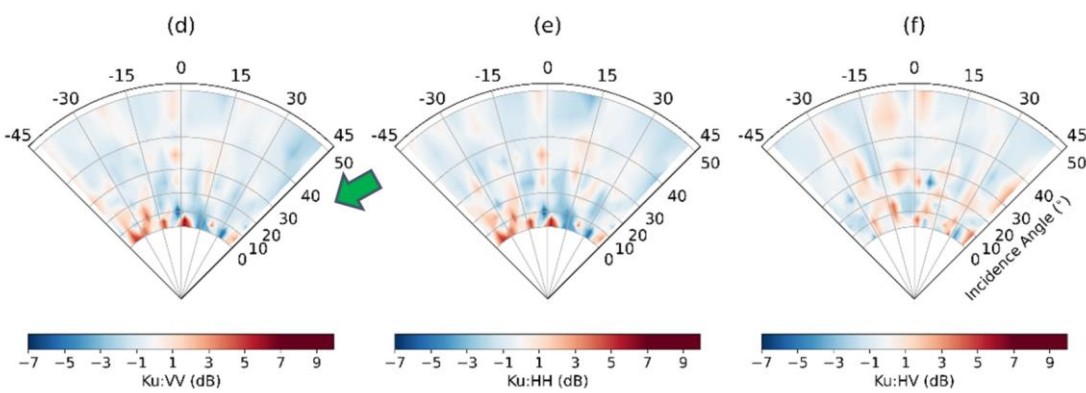


Figure 10: Polar plot panels show the relative change in averaged Ka- and Ku-band backscatter at 5° azimuth sectors, as a function of $\theta_{inc}$, between WE1 and pre-wind conditions, acquired on 11 (WE1) and 9 November, at 2337 UTC and 0013 UTC, respectively. Green arrow denotes the prevailing wind direction on 11 November. The scan times also correspond to yellow circles in Figure 9 and CCTV images in Figure 5a & c. Note: The 11 November CCTV image in Figure 5c is acquired at 1736 UTC for image clarity showing blowing snow.

Compared to azimuthally-averaged Ka- and Ku-band backscatter (Figure 9), spatial variability in Ka- and Ku-band backscatter is evident at all polarizations and $\theta_{inc}$ (Figures 10 and 11) in response to wind events. From pre-wind conditions to WE1, the most striking feature is the development of drifted snow directly in front of the sled (red star in Figure 5) at $\theta_{inc} < 10°$, which led to an increase in Ka- and Ku-band backscatter by up to 9 dB, at nadir throughout all $\theta_{az}$ sectors. Beyond $\theta_{inc} = 10°$, the



change in Ka-band VV and HH backscatter are primarily negative, with spatially heterogeneous areas of positive change,
primarily in the positive $\theta_{az}$ sectors at $\theta_{inc} = 30°$ and $40°$. The change in Ka-band HV backscatter is more consistently positive
at $\theta_{inc} < 10°$ between $0°$ and $-30°$ $\theta_{az}$ sectors, and it agrees well with the strong HV backscatter increase (Figure 9) during the
first snow depositional event that occurred halfway through WE1 on 11 November (Figure 5 and Supplemental Video 1).

WE2 produces a stronger response in Ka- and Ku-band backscatter across the $\theta_{az}$ sectors (Figure 11), compared to WE1. Ka-
band VV and HH backscatter change is primarily negative (up to 7 dB) at $\theta_{inc} > 30°$, while Ka- and Ku-band HV backscatter
shows strong positive change (up to 9.5 dB) at $\theta_{inc} > 40°$. CCTV images (Figures 5d-g) and TLS scans from 8 and 15
November acquired between WE1 and WE2 illustrate changes in surface heights, due to the drifts that formed in the negative
$\theta_{az}$ sectors (purple stars in Figure 5), and this appears to be captured by a strongly enhanced Ku-band HV response (Figure
11f). The large backscatter changes along the negative $\theta_{az}$ sector also indicates change in snow topography from snow blowing
from behind the radar.





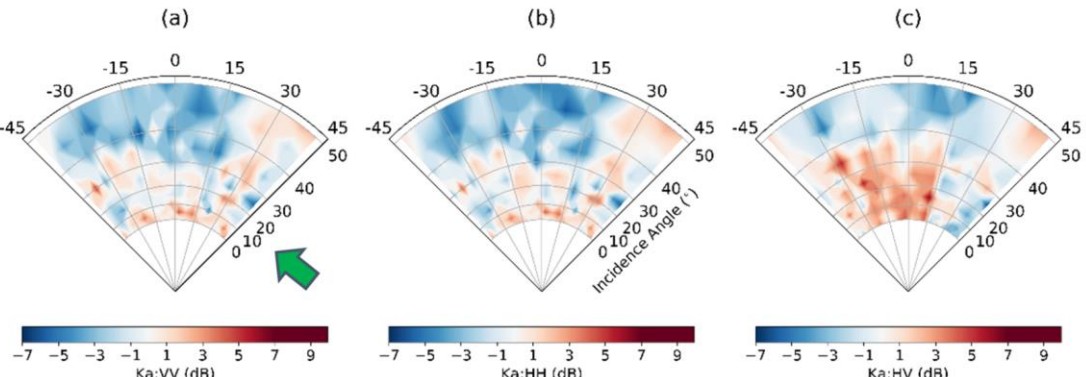

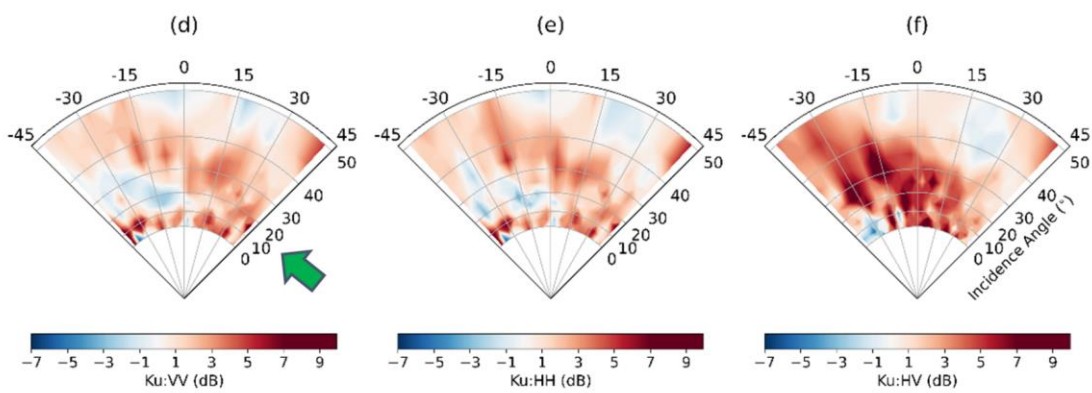

Figure 11: Polar plot panels show the relative change in averaged Ka- and Ku-band backscatter at 5° azimuth sectors, as a function of $\theta_{inc}$, between windy conditions, acquired on 15 (WE2) and 11 (WE1) November, at 2338 UTC and 2337 UTC, respectively. Green arrow denotes the prevailing wind direction on 15 November. The scan times also correspond to yellow circles in Figure 9 and CCTV images in

480                                        Figure 5c & g.

### 3.3.2.2 Co-polarized Phase Difference

Prior to WE1, Ka-band CPD is primarily negative and Ku-band CPD is positive (Figure 12a & b), suggesting stable snow metamorphism during pre-wind conditions. During WE1, Ka- and Ku-band CPD increase from pre-wind conditions at $\theta_{inc} <$ ~ 35°, in positive $\theta_{az}$ sectors (Figure 12c & d). This suggests a short-term wind effect on the snow structure, likely due to

newly deposited snow aligned with the prevailing wind direction (black star in Figure 5e). Also, the horizontal alignment of dunes or newly deposited snow crystals would make new snow layers structurally anisotropic, causing a CPD increase (Leinss et al., 2016). At $\theta_{inc} > 35°$ (Ka-band) and $> 45°$ (Ku-band), the snow located in these sectors appears to have been minimally

affected by the wind (Figures 5a-e and Supplementary Video 1). However, the cold temperatures prior to WE1 (Figure 3c &

d) likely led to significant snow metamorphism in these incidence angle sectors, changing the snow structure alignment from

horizontal towards vertical, causing the CPD to become negative (Leinss et al., 2016).

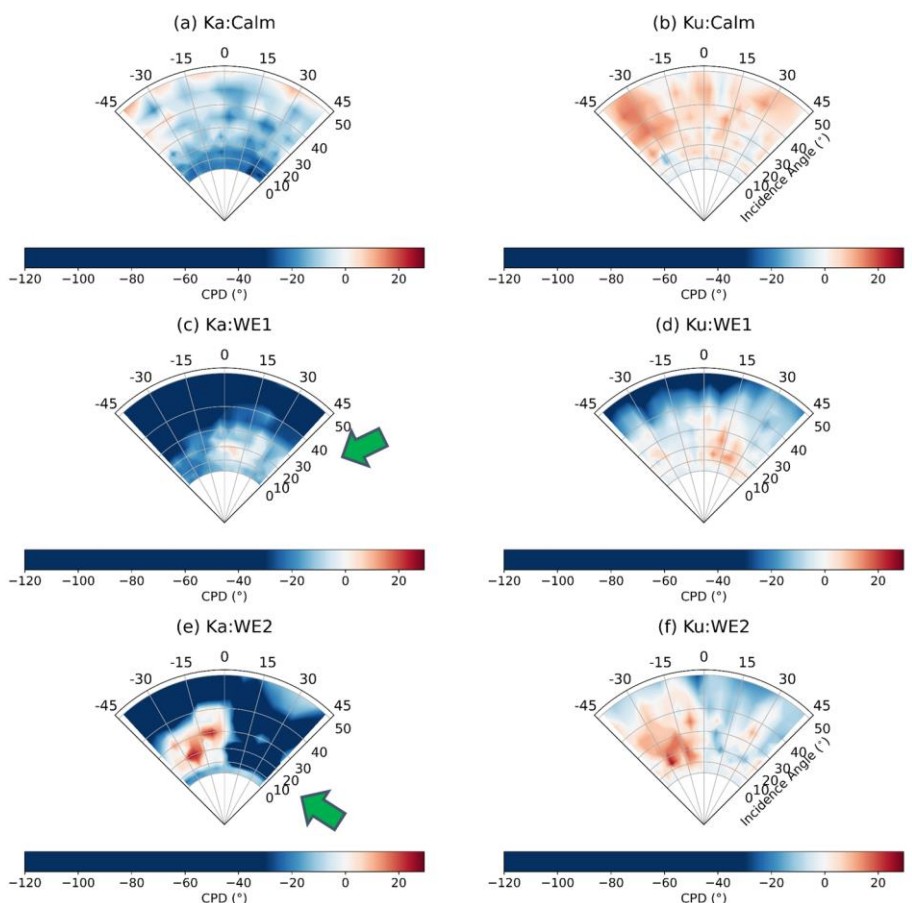

Figure 12: Polar plot panels show averaged Ka- and Ku-band co-polarized phase difference at 5° $\theta_{az}$ sectors, as a function of: (a) & (b)
calm conditions on 9 November (~ 0030 UTC); (c) & (d) WE1 on 11 November (~ 1810 UTC); and (e) & (f) WE2 on 15 November (~
2338 UTC). Green arrow denotes the prevailing wind direction on 11 and 15 November. The scan times also correspond to yellow circles

in Figure 9 and CCTV images in Figure 5.





During WE2, CPD shifts are increasingly negative in the positive $\theta_{az}$ sectors at all $\theta_{inc}$, indicating minimal snow deposition in these sectors during WE2 (Figure 12e & f and Supplemental Video 1). Compared to Ku-band, CPD values are more negative in Ka-band in these sectors, due to its stronger sensitivity to continuous snow metamorphism throughout WE1 and WE2. Compared to WE1, in the negative $\theta_{az}$ sectors, Ka- and Ku-band CPD exhibits phase reversal and stronger positive shift at

$\theta_{inc} < \sim 40°$ (Figure 12e & f). This is likely the result of additional snow redistribution and the resultant formation of two drifts in this sector (purple stars in Figure 5g, and Supplemental Video 1), and with the new snow having horizontal crystal alignment and corresponding phase shift and positive CPD values, stronger at Ka-band.

## 4. Discussion

### 4.1 Impact of Redistributed Snow on Radar Signatures

Our analyses demonstrate that Ka- and Ku-band backscatter and waveforms are sensitive to wind-induced snow redistribution at all polarizations, and incidence angles. During pre-wind conditions, the dominant radar scattering surface switches between the air/snow and snow/sea ice interfaces depending on local variations in the strength of the scattering response between these surfaces. This is shown by the waveform analysis, with the range to the air/snow interface confirmed by georeferencing the radar and TLS data (Figures 7 and 8 and Supplementary Video 2), and the range to the snow/sea ice interface inferred from

local snow depth measurements and the strong interface contrast evident in backscatter in the radar waveforms. Following WE1, the air/snow interface becomes the dominant scattering surface at nadir at all polarizations. With increasing $\theta_{inc}$, the relative scattering contribution of the snow/sea ice interface increases at the expense of the air/snow interface, which gradually becomes invisible (Figure 9). These observations provide contextual information for reliably interpreting backscatter across all polarizations, incidence angles and azimuth ranges.


The Ku- and Ka-band radar backscatter is still sensitive to the presence of buried and historical air/snow interfaces within the snowpack (Figures 7-9), which indicates that snow density and surface roughness contrasts (Figure 4) existing prior to wind events continue to influence scattering even once additional snow is deposited on top (Figure 8). This is an important finding, because even if an interface is not the dominant scattering surface, it can affect the waveform shape and consequently

assumptions about the surface elevation retrieved from airborne and satellite radar altimetry data when there is no *a priori* information on the snow geophysical history.

The relatively small backscatter observed from the snowpack at $\theta_{inc} = 15°$ and $35°$ (Figure 9c & d) indicates that most of the backscatter is associated with the snow/ice interface. This absence of volume scattering change at shallow $\theta_{inc}$, in combination

with the observed nadir sensitivity, suggests that surface scattering is the dominant changing scattering mechanism. The air/snow interface is directly impacted by the wind, experiencing compaction to higher snow density and surface roughness





changes (Figures 4 and 5). The NRCS associated with the air/snow interface changed by more than 5 dB during and following the wind events (Figure 8). Thus, utilizing time-series backscatter at both near- and off-nadir incidence angles may be useful for retrieving snow surface roughness and/or density changes, though it may be difficult to separate these variables.


This study does not replicate airborne- and satellite-scale conditions (e.g., geometry, snow cover and ice type variability on satellite footprint scale, processing such as SAR), due to the experimental setup and footprint size of the KuKa radar. Therefore, the waveform shape, return peak power and measured backscatter from the KuKa radar will be different from airborne and satellite radar altimeters and spaceborne scatterometers or SAR. Also of note is the highly localized KuKa radar backscatter,

which is a two-scale function of microscale surface roughness combined with local $\theta_{inc}$ that includes some steep angles due to snow drifts and bedforms in the footprint. Even at nadir viewing geometry, the beam-limited KuKa radar footprint covers an angular range of 12-17° which is an order of magnitude larger than the beamwidth of a satellite altimeter and larger still than the maximum $\theta_{inc}$ of the altimeter's pulse-limited footprint, which for CryoSat-2 is around 0.1° (Wingham et al., 2006). The dominance of coherent versus non-coherent snow and sea ice backscattering mechanisms can vary significantly between

these incidence angles, with coherent reflections from smooth surfaces dominating the radar response more easily at satellite scales (Fetterer et al., 1992). However, even from a satellite viewing geometry, a rough air/snow interface should produce sufficient backscattering at Ku-band to modify the leading edge of the altimeter waveform response (Landy et al., 2019). The larger satellite footprints may also include undeformed or deformed topography and different scattering surfaces not included in the KuKa radar footprint, such as pressure ridges, rafting and rubble fields, hummocks, smoother refrozen leads, level first-

year sea ice floes and open water. The effects of microscale roughness, larger scale topography and local $\theta_{inc}$ would combine in different ways for larger footprints, such as from satellites operating at large $\theta_{inc}$, where the distribution of local $\theta_{inc}$ may be less extreme (e.g., Segal et al., 2020).

As discussed earlier, the KuKa radar has a much higher vertical resolution than CryoSat-2 (2.5 cm vs 46 cm) and AltiKa (1.5

cm vs 30 cm). This means that although the individual interfaces would not be resolved in the satellite data, the waveform shape and hence retrieved elevation could be affected by current, recent (days), and historical (weeks or longer) timescales of wind-driven redistribution changes to the snow topography and physical properties. Satellite altimetry sea ice retracking algorithms do not yet factor in the potential broadening of the waveform leading edge that could be caused by multiple 'blurred' radar responses from historical buried snow interfaces with a vertical scale smaller than the range resolution of the sensor.

**4.2 Interdependence of Wind and Snow Properties on Backscatter**

This study highlights the influence of snowscape evolution during wind events on backscatter, prompting the need for further investigation of the relative contributions of snow density, surface roughness and snow temperature gradients on Ka- and Ku-band backscatter. There are three main considerations: 1) 'radar-scale' measurement and parameterization of snow surface





roughness are poorly understood, especially its temporal variability; 2) wind induces rapid evolution of snow density (Filhol
& Sturm, 2015); and 3) strong covariance exists between snow temperature, surface density and roughness (Colbeck, 1989).
Although, there is no time series of density profiles available for the RSS, we show a clear increase in density of the upper
snowpack within profiles at comparable locations nearby the RSS (Figure 4). As a snow surface becomes denser, surface
scattering increases due to the enhanced air/snow dielectric contrast. Moreover, as snow becomes warmer, temperature-
gradient driven metamorphism leads to snow surface and volume density changes, which can in turn modify the roughness of
surface and/or internal interfaces, resulting in changes to backscatter (Lacroix et al., 2009).

The waveform analysis does provide some information on the effects of wind vs temperature. In a previous study, the
significant increase in C-band backscatter after a storm was attributed to enhanced radar-scale snow surface roughness and
increasing moisture content in snow with temperatures > -6°C (Komarov et al., 2017). Strong contributions from snow grain
volume scattering at C-band prior to the storm were masked by dominant surface scattering after wind roughening. In our
study, the air and snow surface temperature did not reach -12°C until late on 11 November (Figures 2 and 3), but the increasing
wind speeds during WE1 (Figure 2) were already switching the dominant scattering surface from being a mixture of the
air/snow and snow/ice interface (prior to the wind events), to almost exclusively the air/snow interface, and increasing the
backscatter associated with the air/snow interface by ~ 5 dB (Figure 8). The action of the wind on the snow surface dominated
the change in the scattering surface, and not the increase in air and snow temperature which followed. Therefore, we suggest
the effect of the wind on the snow roughness and/or on the snow density (wind compaction of the top layer) (Figure 4) causes
the air/snow interface to increasingly become the dominant scattering surface at Ka- and Ku-band frequencies.

**4.4 Azimuth Sectoring and Phase Difference**

Azimuth sectoring provides an assessment of the backscatter heterogeneity across the radar footprint, linked to the dynamic
evolution of snow bedforms produced during WE1 and WE2 (Figures 10 and 11). Our results show how sensitive the KuKa
radar is to development of snow bedforms and changing snow surface heights along distinct azimuth sectors within the
footprint with a directionality trend in backscatter, as a function of prevailing wind speed and direction.

Wind-induced snow deposition and snow metamorphism due to high-temperature gradients modified the Ka- and Ku-band
CPD signatures as a function of snow structural anisotropy (Figure 12). This anisotropy induces scale-dependent snow thermal
and dielectric properties (Leinss et al., 2016), further altering the snow surface and interface roughness regimes, and in turn
modifies backscatter and CPD signatures. In general, Ka-band CPD values are higher than Ku-band. At higher frequencies,
more wavelengths fall within the radar wave propagation path length through the snowpack, and the derived CPD becomes
larger (Voglimacci-Stephanopoli et al., 2022; Leinss et al., 2016).

We also observed strong reversals in the CPD following WE2 (Figure 12). CPD reversals could be linked to the wind-
roughening of the air/snow interface during WE2, increasing the chances for multiple scattering/Fresnel reflection in shorter





Ka- and Ku-band wavelengths (Ulaby et al., 1987). The observed phase shift reversals suggest the utility of Ka- and Ku-band CPD to detect and discriminate newly deposited snow and older snow that has undergone temperature gradient metamorphism. Positive phase shifts indicate newly deposited snow (e.g. negative sectors during WE2), while negative phase shifts indicate older/metamorphosed snow (e.g. positive sectors throughout WE1 and WE2). In this study, CPD shifts due to two-way propagation through the snow are not considered because the measured range distances for VV and HH are not significantly different.

## 5. Conclusions

This study details the impact of two wind events on surface-based Ka- and Ku-band radar signatures of snow on Arctic sea ice, collected during the MOSAiC expedition in November 2019. Our results represent the first-ever recording of the impact of snow redistribution on the Ka- and Ku-band radar signatures of snow on sea ice. The formation of snow bedforms and erosion events in the radar footprint modified the snow surface heights as recorded by a terrestrial laser scanner and observable using CCTV imagery. Analysis of radar waveforms demonstrated that the air/snow and snow/sea ice interfaces are visible in both frequencies, all polarisations and incidence angles, and that buried air/snow interfaces remain detectable following new snow deposition. This shows that the historical conditions under which a snow cover evolves, rather than only current conditions, affects backscatter. Hence, historical, and current conditions combine to modify waveform shape and backscatter. We conclude that wind action and its effect on snow density and surface roughness, rather than temperature (which remained < -10°C during the first recorded backscatter shifts), caused the change in the dominant scattering interface from a mixture of air/snow and snow/sea ice interfaces, to predominantly the air/snow interface and nadir backscatter at the air/snow interface increased by up to 5 dB. This effect would likely also be manifest in waveforms detected by satellite altimeters operating at the same frequencies, e.g., AltiKa or CryoSat-2.

Compared to pre-wind conditions, nadir backscatter across the full radar azimuth increased by up to 8 dB (Ka-band) and by up to 5 dB (Ku-band) during the wind events. This was caused by the formation of snow bedforms within the radar footprint, which increased the snow surface roughness and/or density. Azimuth sectoring in 5° bins revealed strong spatial variability in backscatter across the radar footprint. Ka- and Ku-band co-polarised phase difference signatures demonstrate the impact of wind-redistributed snow on phase shifts and its utility to differentiate newly deposited from metamorphosed snow on sea ice. We link this detectability to phase shifts and their dependence on temperature gradient-driven snow metamorphism, and its effect on snow crystal structural anisotropy.

Overall, our results from the KuKa radar provide a process-scale understanding of how wind transport of snow on sea ice affects the dynamic evolution of snow topography and physical properties that influence the accuracy of satellite radar-derived snow depth and sea ice thickness estimates. Our results are relevant to both altimetry and scatterometry through changes to



radar waveforms and backscatter before, during, and after wind events. Our findings however cannot be applied directly to satellite instruments without considering the differences in footprint sizes, incidence angles, and the snow and sea ice properties sampled. However, we provide first-hand information on the frequency, incidence angle and polarisation responses of snow on sea ice, that are vitally important for modelling scattered radiation over an airborne and satellite footprint. With frequent occurrence of winter storms in the Arctic, our findings will facilitate deeper insights and improvements towards better quantifying the impact of snow redistribution to produce accurate retrievals of critical snow/sea ice parameters from presently operational and new high frequency radar altimeter and microwave scatterometer missions such as SARAL/AltiKa, CryoSat-2, Sentinel-3A, Sentinel-6, SWOT, CRISTAL, and ScatSat-1.

*Code and Data availability*: Data used in this manuscript was produced as part of the international Multidisciplinary drifting Observatory for the Study of the Arctic Climate (MOSAiC) with the tag MOSAiC20192020 and the Project ID: AWI PS122 00. KuKa radar data are available at http://data.bas.ac.uk/full-record.php?id=GB/NERC/BAS/PDC/01437. Meteorological data was funded by National Science Foundation (#OPP-1724551).

*Video supplement*. Time-lapse video of wind events acquired by the CCTV camera (Supplemental Video 1) and animation of radar+TLS waveforms during the wind events (Supplemental Video 2) are included in the Appendix.

*Author contributions*. VN processed the KuKa radar data and wrote the manuscript with input from co-authors. RW processed and analyzed the KuKa radar waveforms including NRCS calculations of interfaces, performed TLS data analysis and produced the radar+TLS time series animation. RM processed and plotted the DTC data. DCS processed the TLS data. AJ produced the floe map for the paper. DW and DK processed the SMP data and DW produced the SMP plot. JS, RS, TG and JL provided extensive inputs and reviews to the paper. RT, JY, TN, DJ, MH, JM, GS, SH, RR, MT, MM, MS, DW, MG, CD, IR, CP, IM, and MH provided valuable editorial comments. Many co-authors helped collect data during MOSAiC.

*Competing Interests*: None

*Acknowledgements*. This work was carried out as part of the international Multidisciplinary driftin Observatory for the Study of the Arctic Climate (MOSAiC), MOSAiC20192020 and was funded to JS in part by the Canada 150 Chair program (#G00321321), the National Science Foundation (NSF) Grant #ICER 1928230, and the Natural Environment Research Council (NERC) Grants #NE/S002510/1 & #NE/L002485/1. Funding was also provided to JS by the European Space Agency (grant no. PO #5001027396). VN was additionally supported by Canada's Marine Environmental Observation, Prediction and Response Network (MEOPAR) postdoctoral funds. MG was supported by the DOE Atmospheric System Research Program (#DE−SC0019251, #DE−SC0021341). GS, MH, AJ and SH were supported by the the German Ministry for Education and Research (BMBF) through the MOSAiC IceSense projects (#03F0866B to GS and MH; #03F0866A to AJ and SH) and by the Deutsche Forschungsgemeinschaft (DFG) through the International Research Training Group IRTG 1904 ArcTrain





(#221211316). MS and DW were supported by the European Union's Horizon 2020 research and ARICE (#730965) for MOSAiC berth fees associated with the DEARice participation; and to Swiss Polar Institute grant SnowMOSAiC (#EXF−2018−003). CD was funded through the EUMETSAT MOSAiC project (#4500019119). We thank all scientific personnel and crew members involved in the expedition of the Research Vessel Polarstern during MOSAiC in 2019-2020 (AWI_PS122_00).

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
