# Peer review of "Wind Redistribution of Snow Impacts the Ka- and Ku-band Radar"

_The Cryosphere, 2022_

## Community Comment (CC2)

The authors thank Prof. Andrew Shepherd for his feedback and suggestions. Our detailed responses are given below.

**(1) Title. I find the title to be quite confusing and uninformative; of course wind transport of snow affects radar (and indeed all) signals (sic) over sea ice as it alters the surface height if nothing else! I recommend formulating a title that informs the reader as to what has been found, and not something generic like this.**

Although it is common knowledge that wind alters snow properties, the impact on Ka- and Ku-band radar signals of snow-covered sea ice in both altimetry and scatterometry modes are not well-understood and often neglected in satellite retrieval algorithms. Given our unique opportunity to investigate the influence of wind transport on in situ measured Ka and Ku-band backscatter we chose a title that succinctly summarises our investigation. The word "signature" is used since we investigate the effect on radar waveforms, backscatter, and phase difference. Thus, we disagree with the reviewer that the title is confusing. We however welcome suggestions from the community and will revisit the title when changes are made to the paper during revisions.

**(2). Novelty. It seem from that the data that the authors have observed that increases in snow density (asscoiated with wind transport) lead to reduced volume scattering. This in of itself is not an especially novel conclusion, and so I am wondering whether it is reasonable to claim that this topic is poorly undestood as the authors state in the abstract.**

We show evidence of snow densification and associated increase in snow surface density in the upper layers during the wind events (see Figure 4), and its effect on the radar waveforms and backscatter; an observation that has never been reported during wind events on sea ice. Additionally, we are not aware of any prior observational study into the effect of wind-driven snow redistribution to the contribution of Ka- and Ku-band volume scattering in radar waveforms (lines 80-100) over sea ice. In fact, a large majority of the literature reported in our study have assumed snow redistribution (due to lack of field data) to be a possible factor affecting SAR backscatter and airborne and satellite radar altimeter derived variability in snow depth on sea ice (e.g., Kurtz & Farrell, 2011; Yackel & Barber, 2007). Thus, we argue that this study addresses an important research gap.

Furthermore, for the first time, we combine high-resolution radar and TLS measurements to clearly show both newly deposited and buried snow layers on sea ice modify Ka- and Ku-band radar scattering surfaces (see Figures 6 to 8), backscatter (Figures 9 to 11) and phase difference (Figure 12). All these observations are critical towards accurately modelling radar waveforms and backscatter from radar scattering models such as SMRT. The observations also provide key information to improve retrievals of snow depth and sea ice thickness from satellite radar altimeters on board the AltiKa, CryoSat-2 and forthcoming CRISTAL missions. Therefore, we argue that our findings are novel, fill gaps in our present understanding and can help design future satellite mission algorithms.

**(3) Terminology. I am confused by the use of the term "signatures"; what does this mean? It is implicit, not explicit. Do you mean the radar echoes, or some property of them (e.g. backscattered power., range, etc), or something else?**

The term 'signatures' is a commonly used terminology in radar remote sensing for sea ice applications and encompasses different parameters such as waveforms, derived backscatter, and phase difference (used in this study. It has been used by our community since the earliest radar studies of sea ice (e.g., Rouse, 1969; Livingstone et al., 1987; Rodríguez-Morales et al., 2021). However, we concede that the term could be more explicitly defined in our manuscript, with reference to the parameters we have investigated. We will address this in the revised manuscript.

**(4) Qualitative. As presently written the abstract is almost entirely qualitative, despite there being quite signfiicatn numerical analysis within the paper itself. I recommend using the abstract to summarise the main quantitative conlcusions, which should also support the qualitative conclusions drawn.**

We have carefully constructed the abstract to describe what the salient results are. For example, we describe our main result that the KuKa radar was able to clearly detect buried and new snow layers due to wind events and how this resulted in a shift in the Ka- and Ku-band radar scattering surfaces, the increase in total backscatter and shift in phase difference signatures before, during and after both wind events. We cite relevant text from the abstract below

*"At both frequencies, snow redistribution events increased the dominance of the air/snow interface at nadir as the dominant radar scattering surface, due to wind densifying the snow surface and uppermost layers and modifying the air/snow interface roughness. The radar waveform data also detect the presence of previous air/snow interfaces, buried beneath newly deposited snow. The additional scattering from previous air/snow interfaces could therefore affect the range retrieved from Ka- and Ku-band satellite radar altimeters."*

We also argue that our results are novel due to the one-of-a-kind surface-based radar that mimics key characteristics of satellite radar altimeters. In addition, our results are novel, and they cannot be compared to other results because those directly comparable studies do not exist.

However, we will review the abstract again in the revision phase and refine the abstract in a way that is both qualitatively and quantitatively balanced, keeping the salient results in focus.

**(5) Rigour. Despite collecting a robust and valubale dataset, the authors have stopped short and only report the signal they record rather than complete the analysis to assess the significance of their findings. This leaves the reader to specualte as to whether the findings are in any way important. How much wind is needed to impact on radar data? How are the radar data affected? Is the effect more or less important at Ka or Ku? How does this impact on the scattering horizon, range measurement? How might the effect scale to airborne and satellite measumrents? How typical are the required conditions across the Arctic? There is useful data here, but more work is required to make this a**

**useful contribution to the literature. I recommend that the authors explore the extent to which the changes impact on derived range measurements, for example.**

The authors disagree with the reviewer's usage of 'rigour' as a review subheading and suggest that his comments under that subheading are more relevant to ideas of "impact" than "rigour". We would defend both the impact and rigour of our study.

We performed what we believe to be a rigorous analysis combining several in situ observations, starting from (1) how wind redistributes the snow (TLS – Figure 6) and alters snow properties (temperature, density, SSA from snow pits – Figures 3 and 4), (2) how these snowpack and surface roughness changes impact Ka- and Ku-band radar returns, including waveforms (Figures 7 and 8), total backscatter (Figures 9 and 10) and phase difference (Figures 11 and 12). Additionally, the discussion section has a comprehensive description of the significance of our findings with respect to (a) the wind induced snow volume property changes on radar scattering surfaces, backscatter, and phase difference; supported by TLS and CCTV observations, and (b) validity of our study with respect to scales (surface and satellite), and our focus on the process-scale understanding of how wind affects snow properties and radar returns.

The first wind event (WE1 @12 m/s) on 11 and 12 November 2019 immediately impacted the radar signatures as shown in waveforms and backscatter; and is explained in section 3.2 and discussed in section 4.1. Because this study investigated only two wind events, we are not able to deduce how much wind (i.e., wind speed threshold) is needed to impact radar data. This could be a topic of further investigation.

The authors are happy to agree with the reviewer that more context on how typical wind conditions/thresholds in the Arctic (across space and time) can affect satellite radar returns would be useful. We will add a statement on this in the revised version.

Lines 384-405 describe the wind's impact on Ka- and Ku-band frequencies, separately. We agree that this passage could be improved and will amend it in the revision phase.

With respect to the question on the effect on airborne and satellite measurements, we have mentioned several times in the manuscript that this study is focused on a 'process-scale' understanding of how wind affects the snow cover physical and electromagnetic properties and the impact of those changes on its radar backscatter signatures. For example, the discussion section 4.1 (lines 516-521 and 531-554) acknowledges that this study does not replicate airborne and satellite-scale conditions, but our frequency-, incidence angle- and polarization-dependent results demonstrate the potential of improved algorithms which account for snow redistribution over sea ice to accurately derive snow depth and sea ice thickness (lines 620-624). In the revision phase, we will discuss the practical implementation of our results into retrieval algorithms and evaluation as to whether the increase in computation expense is worth the expected increase in snow depth retrieval accuracy where wind redistribution of snow occurs.

**References**

Rouse, J. W. (1969). Arctic ice type identification by radar. *Proceedings of the IEEE*, *57*(4), 605-611, https://doi.org/10.1109/PROC.1969.7015.

Livingstone, C. E., Onstott, R. G., Arsenault, L. D., Gray, A. L., & Singh, K. P. (1987). Microwave sea-ice signatures near the onset of melt. *IEEE Transactions on Geoscience and Remote Sensing*, (2), 174-187, https://doi.org/10.1109/TGRS.1987.289816.

Rodríguez-Morales, F., Li, J., Alvestegui, D. G. G., Shang, J., Arnold, E. J., Leuschen, C. J., ... & Forsberg, R. (2021). A Compact, Reconfigurable, Multi-UWB Radar for Snow Thickness Evaluation and Altimetry: Development and Field Trials. *IEEE Journal of Selected Topics in Applied Earth Observations and Remote Sensing*, *14*, 6755-676, https://doi.org/10.1109/JSTARS.2021.3092313.

Kurtz, N. T., & Farrell, S. L. (2011). Large-scale surveys of snow depth on Arctic sea ice from Operation IceBridge. *Geophysical Research Letters*, 38(20), https://doi.org/10.1029/2011GL049216.

Yackel, J. J., & Barber, D. G. (2007). Observations of snow water equivalent change on landfast first-year sea ice in winter using synthetic aperture radar data. *IEEE Transactions on Geoscience and Remote Sensing*, 45(4), 1005-1015, https://doi.org/10.1109/TGRS.2006.890418.

---

## Author Comment (AC1)

**Reviewer #1 Comments from Nathan Kurtz**

This is a very interesting and useful study on the impacts of wind-driven changes to Ku and Ka radar returns from ground-based observations during MOSAiC. The study is quite thorough and rigorous, I just had a few minor comments and suggestions for the text as noted below. I would suggest publication subject to some minor revisions.

The authors thank Dr. Kurtz for their valuable time to review our manuscript and suggest publication subject to minor revisions.

Overall, I find the results to be quite interesting to ponder as they show a very detailed look at wind and roughness induced changes in Ku and Ka radar returns. The authors make clear this is a step towards interpreting what factors influence the often-times complex radar returns found in airborne and satellite radar altimeter data, there is not necessarily definitive conclusions to be determined for going from these results to altimeter data but the results are certainly intriguing and worthwhile to publish. I do wonder what this might mean for next steps in terms of future experiments with radar systems on field sites such as this, perhaps this could be added to the end discussion to further highlight what the significance of the data and results may be.

Thanks for your comment. We have added additional concluding statements related to future experiments using the KuKa radar on Arctic and Antarctic sea ice, as follows:

*"In future field-based experiments, we will aim to combine coincident KuKa radar data and terrestrial laser scanner measurements of snow surface roughness to better characterize snow depth retrievals from radar altimetry. Forthcoming KuKa radar deployment campaigns on Antarctic sea ice can further shed valuable insights into complex snow geophysical processes (e.g. slush, presence of melt/refreeze layers, snow-ice formation etc) that may affect snow depth and sea ice thickness retrievals from satellite radar altimetry. With frequent occurrence of winter storms in the Arctic and the Antarctic, our findings will facilitate deeper insights and improvements towards better quantifying the impact of snow redistribution to produce accurate retrievals of critical snow/sea ice parameters from presently operational and new high frequency radar altimeter and microwave scatterometer missions such as SARAL/AltiKa, CryoSat2, Sentinel-3A, Sentinel-6, SWOT, CRISTAL, and ScatSat-1."*

Minor comments

L38: "snow redistribution events increased the dominance of the air/snow interface at nadir as the dominant radar scattering surface" Is the use of the term "dominant" here redundant, or purposeful?

We have changed the sentence to *'At both frequencies, snow redistribution events increased scattering at the air/snow interface at nadir and its prevalence as the dominant radar scattering surface, due to wind densifying the snow surface and uppermost layers.'*

L73-74: I'm not sure the term "originate" is applicable here, perhaps stating they are assumed to be the dominant scattering surface is more appropriate.

Sentence corrected as suggested: *'Under cold and calm snow conditions, a common assumption in radar altimetry is that, the dominant scattering surface of co-polarized Ka- and Ku-band radar signals are assumed to be at the air/snow and snow/sea ice interfaces, respectively (e.g. Lawrence et al., 2018; Tilling et al., 2018), due to dominant surface scattering from these interfaces (Fung & Eom, 1982)'*

Figure 1(a) and (b): What is "foot" in the figures? I think the caption may be describing this, but it would be good if consistent terminology is used.

'Foot' refers to the foot of the radar pedestal that mounts the radar antennas and the positioner system. We have changed the legends to 'distance from pedestal foot' for consistency.

Also L130-131: Is the pedestal here just the platform the radar sits on? Not the phase center of the radar or some determined zero range point?

The radar antennas are mounted on the sides of the positioner, which is mounted on top of the pedestal. The positioner controls the azimuth and incidence angle ranges of the radar system. The radar phase centres are located ~ 16 cm from bottom of Ku antenna (top) and ~ 10 cm from the bottom of the Ka antenna (bottom) as sketched (courtesy: ProSensing Inc) below (not used in the paper).

[Figure]

L128: Isn't the center frequency a bit off from the CryoSat-2 frequency? Or do you mean the frequency ranges overlap?

The frequency ranges are 'close to'. We have modified the sentence as follows:

*'The centre frequency was set to be close to the Ka-band of AltiKa (35 GHz) and the Ku-band of CryoSat-2 (13.575 GHz)'*

L137-138: Given the small range of incidence angles of radar altimeters like CryoSat-2 and AltiKa (mentioned previously in the text), I'm curious what motivated the reasoning

for the 5 degree incidence angle steps? Do you not expect there to be much variation at smaller incidence angles or was this an instrument limitation?

Good question. Having a 5 degree increment to the incidence angle step was not an instrument limitation. The radar was permanently stationed at the remote sensing site along with other radar scatterometers operating at multiple frequencies (C-, X-, L-band etc), acquiring data at the same incidence angle steps and increments similar to the KuKa radar. This was done to acquire consistent measurements of radar scatterometers to collect coincident multi-frequency radar signatures.

Yes, even within incidence angle ranges at shorter angles (e.g. nadir to 5 degrees), we do expect variation in radar backscatter and waveform shapes, however, strongly dependent on the geophysical state of the snowpack, polarization and frequency. In forthcoming campaigns on Arctic and Antarctic sea ice, we are planning to acquire KuKa radar measurements at small incidence angle ranges (e.g. nadir to 5 degrees) at high resolution increments (e.g, 0.5 degrees), to understand the effect of small scale surface roughness changes on radar signatures.

L189-191: What approximate time interval did this integration over the amplitude thresholds typically occur? I think including that could give further evidence to support the fact that you expect the returns to capture the entire snow interval.

The power integration to calculate the NRCS depends on the range of azimuth angles (90 degrees in our case), relative to the beamwidth. The azimuth velocity of the KuKa radar is 5.7 degrees per second. Therefore, for a 90 degree azimuth scan, the time interval for one azimuthal scan across an incidence angle (e.g. nadir) is 15.78 seconds (~ 16 seconds for both frequencies). We have added this information in the revised manuscript as follows:

*The KuKa radar takes ~ 16 seconds (i.e. 5.7° per second) over a 90° $\theta_{az}$ width to acquire data across an incidence angle scan line (e.g. nadir) and ~ 2.5 minutes for one complete scan between $\theta_{inc}$ = nadir to 50°*

L193: A similar statement about the time interval could apply here too. So long as there weren't sea ice ridges or very high surface features in the footprint I think a smaller interval as stated here is appropriate.

The KuKa radar footprint was relatively homogenous and did not contain any ridges or high surface features across the footprint, as observed from the TLS data (see Figure 6) and CCTV imagery (see Figure 5). Line 193 has been modified as follows:

*'The NRCS value is calculated based on the power contained within this peak over an incidence angle scan line, by integrating over the range bins where the power falls below a threshold, are set to -50 dB on either side of the peak for Ka-band data, and -20 dB (-40 dB) on the on the smaller-range (larger-range) sides for Ku-band data.'*

Figure 8: Are these plots from data averaged between -5 to 45 degrees? It seems so from the caption. Is there much difference in plots showing only data from the nadir

direction? I think to some degree this is shown in Figure 9, but I do like the waveform plots in the bottom panel of Figure 8 as a way to show the waveform structure in similar manner to altimetry data.

No. Figures 8 and 9 data are averaged between -25° and +25°, which is the commonly sampled region for Ku- and Ka-band frequencies. We had to correct the KuKa geometry based on the radar system setup. Although the KuKa azimuth scanning range was between -45 and +45 degrees (i.e. 90 degree azimuth range), there is ~ 20 degree offset between the individual antennas and the radar positioner axis origin (see below). The Ku-band antenna therefore scans between ~ -65 degrees to +25 degree azimuth range (region between purple lines) and from -25 degrees to +65 degrees for Ka-band (region between green lines). The region (yellow region between green and purple lines) between -25 degrees and +25 degrees is the overlapping Ku- and Ka-band footprint. We have incorporated these changes in the revised manuscript and in the reviewer #2's rebuttal response.

Yes, the echograms in Figure 9 show data from non-nadir incidence angles. We focused on nadir echograms in Figure 8 to focus more on the altimetry part.

---

## Author Response (AR1)

**Community Comments from Andrew Shepherd**

The authors thank Prof. Andrew Shepherd for his feedback and suggestions. Our detailed responses are given below.

(1) Title. I find the title to be quite confusing and uninformative; of course wind transport of snow affects radar (and indeed all) signals (sic) over sea ice as it alters the surface height if nothing else !  I recommend formulating a title that informs the reader as to what has been found, and not something generic like this.

Although it is common knowledge that wind alters snow properties, the impact on Ka- and Ku-band radar signals of snow-covered sea ice in both altimetry and scatterometry modes are not well-understood and often neglected in satellite retrieval algorithms.  Given our unique opportunity to investigate the influence of wind transport on in situ measured Ka and Ku-band backscatter we chose a title that succinctly summarises our investigation. The word "signature" is used since we investigate the effect on radar waveforms, backscatter and phase difference. Thus, we disagree with the reviewer that the title is confusing. We however welcome suggestions from the community, and will revisit the title when changes are made to the paper during revisions.

(2). Novelty. It seem from that the data that the authors have observed that increases in snow density (asscoiated with wind transport) lead to reduced volume scattering. This in of itself is not an especially novel conclusion, and so I am wondering whether it is reasonable to claim that this topic is poorly undestood as the authors state in the abstract.

We show evidence of snow densification and associated increase in snow surface density in the upper layers during the wind events (see Figure 4), and its effect on the radar waveforms and backscatter; an observation that has never been reported during wind events on sea ice. Additionally, we are not aware of any prior observational study into the effect of wind-driven snow redistribution to the contribution of Ka- and Ku-band volume scattering in radar waveforms (lines 80-100) over sea ice. In fact, a large majority of the literature reported in our study have assumed snow redistribution (due to lack of field data) to be a possible factor affecting SAR backscatter and airborne and satellite radar altimeter derived variability in snow depth on sea ice (e.g. Kurtz & Farrell, 2011; Yackel & Barber, 2007). Thus, we argue that this study addresses an important research gap.

Furthermore, for the first time, we combine high-resolution radar and TLS measurements to clearly show both newly-deposited and buried snow layers on sea ice modify Ka- and Ku-band radar scattering surfaces (see Figures 6 to 8), backscatter (Figures 9 to 11) and phase difference (Figure 12). All these observations are critical towards accurately modelling radar waveforms and backscatter from radar scattering models such as SMRT. The observations also provide key information to improve retrievals of snow depth and sea ice thickness from satellite radar altimeters on board the AltiKa, CryoSat-2 and forthcoming CRISTAL missions. Therefore, we

argue that our findings are novel, fill gaps in our present understanding and can help design future satellite mission algorithms.

(3) Terminology. I am confused by the use of the term "signatures"; what does this mean? It is implicit, not explicit. Do you mean the radar echoes, or some property of them (e.g. backscattered power., range, etc), or something else?

The term 'signatures' is a commonly used terminology in radar remote sensing for sea ice applications and encompasses different parameters such as waveforms, derived backscatter and phase difference (used in this study. It has been used by our community since the earliest radar studies of sea ice (e.g. Rouse, 1969; Livingstone et al., 1987; Rodríguez-Morales et al., 2021). However, we concede that the term could be more explicitly defined in our manuscript, with reference to the parameters we have investigated. We will address this in the revised manuscript.

(4) Qualitative. As presently written the abstract is almost entirely qualitative, despite there being quite signfiicatn numerical analysis within the paper itself. I recommend using the abstract to summarise the main quantitative conlcusions, which should also support the qualitative conclusions drawn.

We have carefully constructed the abstract to describe what the salient results are. For example, we describe our main result that the KuKa radar was able to clearly detect buried and new snow layers due to wind events and how this resulted in a shift in the Ka- and Ku-band radar scattering surfaces, the increase in total backscatter and shift in phase difference signatures before, during and after both wind events. We cite relevant text from the abstract below

*"At both frequencies, redistribution caused snow densification at the surface and the uppermost layers, increasing the scattering at the air/snow interface at nadir and its prevalence as the dominant radar scattering surface. The waveform data also detected the presence of previous air/snow interfaces, buried beneath newly deposited snow. The additional scattering from previous air/snow interfaces could therefore affect the range retrieved from Ka- and Ku-band satellite altimeters."*

We also argue that our results are novel due to the one-of-a-kind surface-based radar that mimics key characteristics of satellite radar altimeters. In addition, our results are novel and they cannot be compared to other results because those directly comparable studies do not exist.

However, we will review the abstract again in the revision phase and refine the abstract in a way that is both qualitatively and quantitatively balanced, keeping the salient results in focus.

(5) Rigour. Despite collecting a robust and valubale dataset, the authors have stopped short and only report the signal they record rather than complete the analysis to assess the significance of their findings. This leaves the reader to specualte as to whether the findings are in any way important. How much wind is needed to impact on radar data? How are the radar data affected? Is the effect more or less important at Ka or Ku? How does this impact on the scattering horizon, range measurement? How might the effect scale to airborne and satellite measumrents? How

typical are the required conditions across the Arctic? There is useful data here, but more work is required to make this a useful contribution to the literature. I recommend that the authors explore the extent to which the changes impact on derived range measurements, for example.

The authors disagree with the reviewer's usage of 'rigour' as a review subheading and suggest that his comments under that subheading are more relevant to ideas of "impact" than "rigour". We would defend both the impact and rigour of our study.

We performed what we believe to be a rigorous analysis combining several in situ observations, starting from (1) how wind redistributes the snow (TLS – Figure 6) and alters snow properties (temperature, density, SSA from snow pits – Figures 3 and 4), (2) how these snowpack and surface roughness changes impact Ka- and Ku-band radar returns, including waveforms (Figures 7 and 8) and total backscatter (Figures 9 and 10). Additionally, the discussion section has a comprehensive description of the significance of our findings with respect to (a) the wind induced snow volume property changes on radar scattering surfaces and backscatter; supported by TLS and CCTV observations, and (b) validity of our study with respect to scales (surface and satellite), and our focus on the process-scale understanding of how wind affects snow properties and radar returns.

The first wind event (WE1 @12 m/s) on 11 and 12 November 2019 immediately impacted the radar signatures as shown in waveforms and backscatter; and is explained in section 3.2 and discussed in section 4.1. Because this study investigated only two wind events, we are not able to deduce how much wind (i.e. wind speed threshold) is needed to impact radar data. This could be a topic of further investigation. We have added a sentence in the conclusion as follows:

*'Our results are relevant to both satellite altimetry and scatterometry through changes to radar waveforms and backscatter during, and after wind events. However, more investigation is needed to deduce how much wind (i.e., conditions/thresholds across space and time) is needed to impact satellite waveforms.'*

The authors are happy to agree with the reviewer that more context on how typical wind conditions/thresholds in the Arctic (across space and time) can affect satellite radar returns would be useful. We have added a sentence in the conclusion as above

Lines 384-405 describe the wind's impact on Ka- and Ku-band frequencies, separately. We agree that this passage could be improved and will amend it in the revision phase.

With respect to the question on the effect on airborne and satellite measurements, we have mentioned several times in the manuscript that this study is focused on a 'process-scale' understanding of how wind affects the snow cover physical and electromagnetic properties and the impact of those changes on its radar backscatter signatures. For example, the discussion section 4.1 (lines 516-521 and 531-554) acknowledges that this study does not replicate airborne and satellite-scale conditions, but our frequency-, incidence angle- and polarization-dependent results demonstrate the potential of improved algorithms which account for snow

redistribution over sea ice to accurately derive snow depth and sea ice thickness (lines 620-624).

A recent study by Nab et al. (2023) shows a vertical shift in CryoSat-2 derived radar freeboard due to snow accumulation, higher wind speeds and warmer air temperatures; similar to what we observe from our surface-based observations. In the revised version, we have discussed this in detail as follows:

*'At satellite scales, this may upwardly shift the retracked elevation and resulting sea ice freeboard retrievals by radar altimeters that assume the snow/sea ice interface is the dominant scattering surface. This would introduce an overestimating bias on the sea ice thickness estimate; however, a number of other uncertainties are also at play in this process, meaning this may move the retrieval closer or further from the true value. Our surface-based findings are consistent with recent satellite-based work by Nab et al. (2023), who showed a temporary lifting of CryoSat-2-derived radar freeboard in response to snow accumulation, but also higher wind speeds and warmer air temperatures.'*

With regards to quantifying the extent of range changes during the wind events, we do not have snow depth measurements from the scan area limiting our ability to validate the snow depth measurements from the scan area vs KuKa-derived snow depth; and not the focus of this paper. The local snow depth measurements are only from behind the radar and directly in front of the radar (estimated based on sled height). We already describe this in section 3.2 (Lines 307-317) in the revised manuscript as follows:

*'Prior to WE1, radar waveforms remained stable, with only small power variations over time. The peak power at VV and HH generally corresponds to the air/snow interface in most bins, as also confirmed by the TLS-derived heights. A lower scattering interface is also visible at ~ 20 to 40 cm below the air/snow interface, especially prominent in the HV data in both frequencies, but also visible in the VV and HH data. The range values indicated in the radar waveforms are based on the speed of light in free space. Correcting for a reduction of 80% for snow (Willatt et al., 2009), the lower interfaces lay ~ 16 to 32 cm below the air/snow interface. To better understand this, we consider the HV waveform and local snow depth. Snow depth measured behind the scan area during 4 and 14 November varied between 21 and 29 cm. Based on the very small amount of radiation scattered from larger ranges, negligible penetration of Ku- and Ka-band signals into sea ice (Fung et al., 1994), and the consistency with local snow depth, this interface in the HV data is very likely the snow/ice interface. A small amount of returned power is expected from beyond due to snow and ice backscattering from the perimeter of the 30-50 cm radar scan area and sidelobes.'*

The ideal way to quantify the extent of range changes during wind events is to coincidentally acquire radar scans and sample snow depth directly in front of the radar (i.e. destructive sampling) or to use the stare mode of KuKa radar in transects where the radar is towed along a transect and snow depth is near-coincidentally measured before or after a radar measurement (see Stroeve et al., 2020). We have added a line in the conclusion as follows:

*'In future field-based experiments, we will aim to combine near-coincident KuKa radar data and snow depth measurements (Stroeve et al., 2020), terrestrial laser scanner measurements of snow surface roughness and snow density profiles to better characterise the effect of these variables on the radar range measurements.'*

**References**

Rouse, J. W. (1969). Arctic ice type identification by radar. *Proceedings of the IEEE*, *57*(4), 605-611, https://doi.org/10.1109/PROC.1969.7015.

Livingstone, C. E., Onstott, R. G., Arsenault, L. D., Gray, A. L., & Singh, K. P. (1987). Microwave sea-ice signatures near the onset of melt. *IEEE Transactions on Geoscience and Remote Sensing*, (2), 174-187, https://doi.org/10.1109/TGRS.1987.289816.

Rodríguez-Morales, F., Li, J., Alvestegui, D. G. G., Shang, J., Arnold, E. J., Leuschen, C. J., ... & Forsberg, R. (2021). A Compact, Reconfigurable, Multi-UWB Radar for Snow Thickness Evaluation and Altimetry: Development and Field Trials. *IEEE Journal of Selected Topics in Applied Earth Observations and Remote Sensing*, *14*, 6755-676, https://doi.org/10.1109/JSTARS.2021.3092313.

Kurtz, N. T., & Farrell, S. L. (2011). Large-scale surveys of snow depth on Arctic sea ice from Operation IceBridge. *Geophysical Research Letters*, 38(20), https://doi.org/10.1029/2011GL049216.

Yackel, J. J., & Barber, D. G. (2007). Observations of snow water equivalent change on landfast first-year sea ice in winter using synthetic aperture radar data. *IEEE Transactions on Geoscience and Remote Sensing*, 45(4), 1005-1015, https://doi.org/10.1109/TGRS.2006.890418.

Nab, C., Mallett, R., Gregory, W., Landy, J., Lawrence, I., Willatt, R., ... & Tsamados, M. (2023). Synoptic variability in satellite altimeter-derived radar freeboard of Arctic sea ice. Geophysical Research Letters, e2022GL100696, https://doi.org/10.1029/2022GL100696

Stroeve, J., Nandan, V., Willatt, R., Tonboe, R., Hendricks, S., Ricker, R., ... & Tsamados, M. (2020). Surface-based Ku-and Ka-band polarimetric radar for sea ice studies. *The Cryosphere*, 14(12), 4405-4426, https://doi.org/10.5194/tc-14-4405-2020.

**Reviewer #1 Comments from Nathan Kurtz**

This is a very interesting and useful study on the impacts of wind-driven changes to Ku and Ka radar returns from ground-based observations during MOSAiC. The study is quite thorough and rigorous, I just had a few minor comments and suggestions for the text as noted below. I would suggest publication subject to some minor revisions.

The authors thank Dr. Kurtz for their valuable time to review our manuscript and suggest publication subject to minor revisions.

Overall, I find the results to be quite interesting to ponder as they show a very detailed look at wind and roughness induced changes in Ku and Ka radar returns. The authors make clear this is a step towards interpreting what factors influence the often-times complex radar returns found in airborne and satellite radar altimeter data, there is not necessarily definitive conclusions to be determined for going from these results to altimeter data but the results are certainly intriguing and worthwhile to publish. I do wonder what this might mean for next steps in terms of future experiments with radar systems on field sites such as this, perhaps this could be added to the end discussion to further highlight what the significance of the data and results may be.

Thanks for your comment. We have added additional concluding statements related to future experiments using the KuKa radar on Arctic and Antarctic sea ice, as follows:

*"In future field-based experiments, we will aim to combine coincident KuKa radar data, snow depth, terrestrial laser scanner measurements of snow surface roughness and snow density profiles to better characterise the effect of these variables on the radar range measurements. Forthcoming KuKa radar deployments on Antarctic sea ice will produce further insights into snow geophysical processes (e.g. presence of slush, melt/refreeze layers, snow-ice formation etc.) that may affect snow depth and sea ice thickness retrievals from satellite radar altimetry. In a windy Arctic and the Antarctic, these methods will facilitate improved insights towards better quantifying the impact of snow redistribution on accurate retrievals of snow/sea ice parameters from satellite radar missions such as SARAL/AltiKa, CryoSat2, Sentinel-3A, Sentinel-6, SWOT, CRISTAL, and ScatSat-1."*

Minor comments

L38: "snow redistribution events increased the dominance of the air/snow interface at nadir as the dominant radar scattering surface" Is the use of the term "dominant" here redundant, or purposeful?

We have changed the sentence to *'At both frequencies, redistribution caused snow densification at the surface and the uppermost layers, increasing the scattering at the air/snow interface at nadir and its prevalence as the dominant radar scattering surface.'*

L73-74: I'm not sure the term "originate" is applicable here, perhaps stating they are assumed to be the dominant scattering surface is more appropriate.

Sentence corrected as suggested: *'Under cold snow conditions, a common assumption in radar altimetry is that the dominant scattering surfaces of co-polarized Ka- and Ku-band radar signals correspond to the air/snow and snow/sea ice interfaces, respectively (e.g. Armitage et al., 2015; Tilling et al., 2018).'*

Figure 1(a) and (b): What is "foot" in the figures? I think the caption may be describing this, but it would be good if consistent terminology is used.

'Foot' refers to the foot of the radar pedestal that mounts the radar antennas and the positioner system. We have changed the legends to 'distance from pedestal foot' for consistency.

Also L130-131: Is the pedestal here just the platform the radar sits on? Not the phase center of the radar or some determined zero range point?

The radar antennas are mounted on the sides of the positioner, which is mounted on top of the pedestal. The positioner controls the azimuth and incidence angle ranges of the radar system. The radar phase centres are located ~ 16 cm from bottom of Ku antenna (top) and ~ 10 cm from the bottom of the Ka antenna (bottom) as sketched (courtesy: ProSensing Inc) below (not used in the paper).

[Figure]

L128: Isn't the center frequency a bit off from the CryoSat-2 frequency? Or do you mean the frequency ranges overlap?

The frequency ranges are 'close to'. We have modified the sentence as follows:

*'The central frequency of the radar chirps were set to be close to the Ka-band of AltiKa (35 GHz) and the Ku-band of CryoSat-2 (13.575 GHz).'*

L137-138: Given the small range of incidence angles of radar altimeters like CryoSat-2 and AltiKa (mentioned previously in the text), I'm curious what motivated the reasoning for the 5 degree incidence angle steps? Do you not expect there to be much variation at smaller incidence angles or was this an instrument limitation?

Good question. Having a 5 degree increment to the incidence angle step was not an instrument limitation. The radar was permanently stationed at the remote sensing site along with other radar scatterometers operating at multiple frequencies (C-, X-, L-band etc), acquiring data at the same incidence angle steps and increments similar to the KuKa radar. This was done to acquire consistent measurements of radar scatterometers to collect coincident multi-frequency radar signatures.

Yes, even within incidence angle ranges at shorter angles (e.g. nadir to 5 degrees), we do expect variation in radar backscatter and waveform shapes, however, strongly dependent on the geophysical state of the snowpack, polarization and frequency. In forthcoming campaigns on Arctic and Antarctic sea ice, we are planning to acquire KuKa radar measurements at small incidence angle ranges (e.g. nadir to 5 degrees) at high resolution increments (e.g, 0.5 degrees), to understand the effect of small scale surface roughness changes on radar signatures.

L189-191: What approximate time interval did this integration over the amplitude thresholds typically occur? I think including that could give further evidence to support the fact that you expect the returns to capture the entire snow interval.

The power integration to calculate the NRCS depends on the range of azimuth angles (90 degrees in our case), relative to the beamwidth. The azimuth velocity of the KuKa radar is 5.7 degrees per second. Therefore, for a 90 degree azimuth scan, the time interval for one azimuthal scan across an incidence angle (e.g. nadir) is 15.78 seconds (~ 16 seconds for both frequencies). We have added this information in the revised manuscript as follows:

*The KuKa radar takes ~ 16 seconds (i.e. 5.7° per second) over a 90° width to acquire data across an incidence angle scan line and ~ 2.5 minutes for one complete scan between = 0° - 50°.*

L193: A similar statement about the time interval could apply here too. So long as there weren't sea ice ridges or very high surface features in the footprint I think a smaller interval as stated here is appropriate.

The KuKa radar footprint was relatively homogenous and did not contain any ridges or high surface features across the footprint, as observed from the TLS data (see Figure 6) and CCTV imagery (see Figure 5). Line 193 has been modified as follows:

*'For waveform analysis, we calculated the NRCS values at nadir for the air/snow and snow/ice interfaces by integrating the power over the waveform peaks within +/- 2 dB either side from the overlapping scan area (Section 3.2). Next, we calculated the NRCS value integrated over the entire snow volume based on the power contained within this peak over an incidence angle scan line, by integrating over the range bins where the power falls below a threshold, set to -50 dB on either side of the peak for Ka-band data, and -20 dB (-40 dB) on the on the smaller-range (larger-range) sides for Ku-band data.'*

Figure 8: Are these plots from data averaged between -5 to 45 degrees? It seems so from the caption. Is there much difference in plots showing only data from the nadir direction? I think

to some degree this is shown in Figure 9, but I do like the waveform plots in the bottom panel of Figure 8 as a way to show the waveform structure in similar manner to altimetry data.

No. Figures 8 and 9 data are averaged between -25° and +25°, which is the commonly sampled region for Ku- and Ka-band frequencies. We had to correct the KuKa geometry based on the radar system setup. Although the KuKa azimuth scanning range was between -45 and +45 degrees (i.e. 90 degree azimuth range), there is ~ 20 degree offset between the individual antennas and the radar positioner axis origin (see below). The Ku-band antenna therefore scans between ~ -65 degrees to +25 degree azimuth range (region between purple lines) and from -25 degrees to +65 degrees for Ka-band (region between green lines). The region (yellow region between green and purple lines) between -25 degrees and +25 degrees is the overlapping Ku- and Ka-band footprint. We have incorporated these changes in the revised manuscript and in the reviewer #2's rebuttal response.

Yes, the echograms in Figure 9 show data from non-nadir incidence angles. We focused on nadir echograms in Figure 8 to focus more on the altimetry part.

[Figure]

[Figure]

**Reviewer #2 Comments from Silvan Leinss**

Review, "Wind Transport of Snow Impacts Ka- and Ku-band Radar Signatures on Arctic Sea Ice" by V. Nandan et al.

The manuscript presents waveform and backscatter time series aquired over snow covered sea ice with a ground-based radar altimeter and scatterometer. The instrument has an exceptional bandwidth of 6 GHz at Ku-band (13.6 GHz) and 10 GHz at Ka-band (35 GHz) resulting in centimeter slant-range resolution. The instrument can provide detailed and relevant information to interpret data from existing and future radar altimeter missions like CryoSat-2, CRISTAL, SARAL, etc. Publishing such data and time series is a valuable contribution suitable for the journal The Cryosphere (TC). The manuscript, however, requires thoroughly revision (major) to meet scientific publishing standards and to focus more concisely on the most relevant aspects of the valuable work.

The most relevant points of manuscript are:

- Very few ground-based radar data especially over sea ice exists. The manuscript contains detailed plots of such data which provides valuable observations.

 - The time series plots of very high (centimeter) resolution Nadir backscatter returns reveal the dynamics of the relative scattering contribution of the main interfaces air/snow and snow/ice. The data show that air/snow interface is not necessarily the strongest scattering contribution , even at Ka-band, which is valuable for future radar altimeter missions.

- during pre-wind conditions the dominant radar return (at 13 and 35 GHz) at nadir alternates between the air/snow interface and the snow/ice interface.

 - The creation of hard Wind slabs causes an increase in backscatter at Nadir and a decrease in backscatter for non-nadir angles (observed at 15...50°).

 - The comparison of Nadir radar returns with off-nadir (theta_inc > 15°) is a valuable information regarding penetration depth of radar sounders/altimeters (less penetration) vs. SAR systems (larger penetration due to slant-looking gemometry).

The authors thank Dr. Leinss for their valuable time to review and provide feedback to improve our manuscript. Following is our rebuttal (in black) to reviewer's comments (in red) and associated changes in the revised version of the manuscript.

**General comments:**

The manuscript needs considerable improvements in readability and focus.

 - Try to focus/condense on most relevant aspects (see above),

We agree with your comment on condensing the main points. The revised version now focuses only on the ranging and backscatter analysis, supported by TLS, meteorological and geophysical observations.

a) We have revised the KuKa radar geometry section with detailed schematic of the footprint and measurements (see section 2.1 and revised figure 1 in the revised manuscript)
b) We agree that the phase difference analysis distracts from the main message of this paper and requires more analysis for better interpretation. Therefore, we have removed the phase difference analysis from this paper.
c) To avoid confusion, we have removed the concept of positive and negative azimuth sectoring description. Instead, we focus on individual footprint azimuth angles for Ka- and Ku-band frequencies (see last paragraph of section 2.5).
d) Points a) and c) has led to revised interpretation of the backscatter analysis, supported by TLS, radar ranging and snow geophysical data.

- consider to shorten where possible, e.g. the polar plots seem not to reveal too much information;

We do agree the irrelevance of co-pol phase difference analysis in this paper via polar plots and we have removed the entire analysis from the paper. However, we have retained the backscatter analysis using azimuth sectoring as the polar plots clearly show spatial variability in backscatter within the footprint, as a function of incidence angle and polarization, in response to wind-driven snow dynamics and thermodynamic-induced metamorphic changes to snow layering. This analysis distinctly shows the importance of accounting for spatiotemporal variability in snow properties within a satellite footprint which can contribute to the overall measured changes in dominant scattering horizon and backscatter.

In the revised version, we have revised this analysis through improved representation of KuKa radar geometry (see changes and depiction below in the rebuttal).

- results/discussion/conclusion: check for redundant, irrelevant information.

The authors agree with your comment. We have removed the CPD analysis from the paper and discussed only the relevant information related to our primary objectives (ranging and backscatter analysis). We have also revised our interpretation on azimuth sectoring (section 3.3.2) based on changes in the KuKa radar geometry and removal of the 'positive' and 'negative' azimuth sector concept. In the radar backscatter analysis (section 3.3.1), we have refined our interpretations, particularly with respect to non-nadir incidence angles at 15 and 35 degrees.

- Give the main points more weight by improving the graphics showing these results.

We have modified and improved the plots as suggested for Figures 1, 2, 3, 6, 8 and merged Figures 9 and 10 into one single (now Figure 9 based on the revised KuKa radar geometry).

- I really don't understand what is the exact extend and overlap of the observed area with Ku- and Ka band. Different ranges of azimuth angles are given in the text and figures; please clarify this using an observation-setup-sketch showing all relevant geometric parameters.

The reviewer makes a valuable point here. We reviewed the KuKa footprint geometry based on the radar system setup. Although the KuKa azimuth scanning range was set between -45 and +45 degrees (i.e. 90 degree azimuth range), there is ~ 20 degree offset between the individual antennas and the radar positioner axis origin (see revised Figure 1 below). The Ku-band antenna therefore scans between ~ -65 degrees to +25 degree azimuth range (region between purple lines) and from -25 degrees to +65 degrees for Ka-band (region between green lines) (Figure 1(b), (e) and (f)). The region (yellow) between green and purple lines) between -25 degrees and +25 degrees is the overlapping Ku- and Ka-band footprint (Figure 1(b)).

[Figure]

*Figure 1: KuKa radar geometry illustrating (a) radial distance and radar range from the pedestal foot; (b) KuKa radar azimuth scan pattern projected based on the positioner axis coordinate system; (c) diameter of radar footprint, measured radially ('ra') and azimuthally ('az'); and (d) area of radar footprint; (e) and (f) depicts the Ku- and Ka-band footprints of the KuKa radar, respectively. In panel (b), the region between purple lines and green lines are the respective Ku- and Ka-band footprints (separately shown in panels (e) and (f)), while the yellow region in (b) is the overlapping footprint area for both frequencies.*

The revised figures 8-11 are averaged between -25° and +25°, which is the commonly sampled region for Ku- and Ka-band frequencies. In section 2.1, we have provided a detailed description of the overlapping KuKa geometry, as follows:

*"During MOSAiC, the KuKa radar scanned over a 90° continuous  range width for every 5° interval in . The KuKa radar takes ~ 16 seconds (i.e. 5.7° per second) over a 90°  width to acquire data across an incidence angle scan line and ~ 2.5 minutes for one complete scan between  = 0° - 50°. However, there is a ~ 20° offset between the individual radar antennas and the radar positioner axis origin. Therefore, the Ku-band antenna scans between -65° to +25°  range (region between purple lines) from -25° to +65° for Ka-band (region between green lines) (Figure 1(b), (e) and (f)). This also means that the Ku- and Ka-band scan area overlap for a given radar 'shot' is  dependent. The yellow region between green and purple lines in Figure 1b between -25° and +25° is the overlapping Ku- and Ka-band scan area."*

 - Try to provide more convincing conclusions where possibly and try to reduce speculative interpretations throughout the manuscript.

We agree with your comment on condensing the main points. The revised version now focuses only on the ranging and backscatter analysis, supported by TLS, meteorological and geophysical observations. We have removed the phase difference analysis from this paper as we agree it is not relevant to the main message of this paper.

 - There are several interpretations/speculations related to surface roughness: consider removing them where not really necessary.

We recognise that, though we acknowledge the potential importance of roughness, we do not have quantitative data to investigate its effect. We therefore retain mention of roughness in the manuscript but try to explain this more clearly – we have added several sentences in the second paragraph of the Discussion (section 4.1) to clarify that this is probably important, but not collected in this study, and the need to collect high-temporal resolution surface roughness measurements is paramount (see below).

*'In future studies, gathering TLS data on the snow surface roughness at high spatial (radar) and temporal (e.g., daily or hourly) resolution would provide valuable information on the role of roughness. In addition, collecting near-coincident measurements of snow density would provide information on the role of density affecting radar waveforms. We would therefore recommend collecting these coincident datasets in future similar studies.'*

**Specific comments:**

35/36: It would be good to mention in the abstract what kind of radar is used and for which observation which mode was used: scatterometer or altimeter or both modes of the instrument?

We have revised the text as follows: *"Here, we examine the effects of snow redistribution over Arctic sea ice on radar waveforms and backscatter signatures obtained from a surface-based, fully-polarimetric Ka- and Ku-band radar, at incidence angles between 0° (nadir) and 50°."*

37: "waveforms and backscatter": I guess, they were measured in altimeter mode? Please mention.

We have revised the lines as suggested as follows: "*During both events, changes in Ka- and Ku-band radar waveforms and backscatter coefficients at nadir are observed, coincident with surface height changes measured from a terrestrial laser scanner*"

39/40: "...altimeter. The relative ... decreases, ... with increasing incidence angle." For the described observations, it seems like the scatterometer mode was used here? The sentence before ends with altimeter; this sentence mentions "increasing incidence angle" which does not match with altimeter. Please clarify; maybe start with "With increasing incidence angle of the scatterometer, the relative scattering contribution ...  ..." if that agrees with what you mean.

To clarify, the KuKa radar operates in altimeter and scatterometer modes. In the altimeter mode, the radar acquires data ONLY at nadir, while in the scatterometer mode, the radar can also acquire data at large incidence angles (including nadir). In this study, we use the scatterometer mode data from nadir to 50° incidence angles. We have avoided terming *altimeter* and *scatterometer* modes in this paper as it slightly conflicts with the instrument terminology for the radar.

We agree with your comment on lines 39-40 and is revised as follows: "*With increasing incidence angles, the relative scattering contribution of the air/snow interface decreases, and the snow/sea ice interface scattering increases.*"

124 and Fig. 1b: It would be good to

 - 1) refer/use Fig. 1b to better explain/clarify how the "azimuth range" is scanned, at discrete incidence angles. I guess, Fig 1b shows the azimuth-elevation scan pattern of the instrument;

In addition to the revised Figure 1b, we have added information regarding this comment, as follows:

"*During MOSAiC, the KuKa radar scanned over a 90° continuous  range width for every 5° interval in . The KuKa radar takes ~ 16 seconds (i.e. 5.7° per second) over a 90°  width to acquire data across an incidence angle scan line and ~ 2.5 minutes for one complete scan between  = 0° - 50°.*"

 - 2) this could be mentioned in the caption and as an axis-label on Fig. 1b (at least azimuth).

We have instead provided this information in the main text, as above.

132: How is the beam-width defined? full-width-at-half-maximum (FWHM) equivalent to the "3dB-beam width"? Strove 2020 writes "Antenna 6 dB two-way beamwidth".

Each radar antenna system has a 3 dB half-power beam width of 6°. We have added this information into the manuscript as follows: '*The antenna beamwidth (6 dB two-way) is 16.9°*

*and 11.9° for Ku- and Ka- bands, respectively. Therefore, the size of the radar scan area on the snow is dependent on frequency, height of the antenna above the snow surface, and $\theta_{inc}$.*

135: "The overlapping footprint is between -5 and +45° for Ku-band, and -45 to +5° for Ka-band": I don't understand this sentence. To me, this seems to contradict the sentence before and the geometric radar setup as shown by Strove 2020. The overlap in % (as given by Strove) of course depends on incidence angle, but why is the azimuth angle mentioned here? Why is the "overlapping footprint" for Ku-band at positive angles and at negative az-angles for Ka-band?

We have revised the description of the scan geometry and the overlapping footprint. The overlapping footprint is between -25 and +25 degrees based on geometry calculation as shown in Figure 1(b) (yellow region). The revised description is as follows:

*However, there is a ~ 20° offset between the individual radar antennas and the radar positioner axis origin. Therefore, the Ku-band antenna scans between -65° to +25° range (region between purple lines) from -25° to +65° for Ka-band (region between green lines) (Figure 1(b), (e) and (f)). This also means that the Ku- and Ka-band scan area overlap for a given radar 'shot' is dependent. The yellow region between green and purple lines in Figure 1b between -25° and +25° is the overlapping Ku- and Ka-band scan area.*

Related to that, I don't understand the yellow-orange-red color in Figure 6;

We agree it was not an ideal choice of colours. We have re-coloured Figure 6 following Figure 1, and noted this in the figure caption below

[Figure]

*'Figure 6: TLS data (plan view) from 1, 8 and 15 November, from -90° to + 90°, where the angle indicates the azimuth of the radar positioner, and radial horizontal distance measured from the centre of the radar pedestal. The top panels show the topography as measured*

*downwards (increasing negative) from the middle of the radar antenna arms. Black indicates no data recordings in that bin. Projections of the centres of the radar footprints are shown for 0° and 50° radar inclination angles, superimposed on the TLS data in magenta and green for radar observations, respectively, and buff where the two overlap, as per Figure 1. The bottom panels indicate the number of TLS data points within each bin. Surface depressions resulting in 0 counts in the TLS data are due to obscuration by adjacent high areas due to snow/sea ice topography and human-made objects, as viewed from the TLS's oblique viewpoint some distance away.'*

I don't understand why Figure 7 shows az-angles of -65..+25° for Ku-band and -25° to +65 for Ka band;

These angles are the azimuth angle ranges for Ku- and Ka-bands, respectively. To avoid confusion, we have revised the portion of the figure caption as follows:

*Figure 7: Progression of Ka- and Ku-band radar waveforms at nadir between -65° to +25° (Ku-band) and -25° to +65° (Ka-band) (azimuth ranges following Figure 1(e) and (f)).*

I don't understand why Figure 10-12 show azimuth angles for -45 to +45 for both Ku and Ka band.

We have replotted the polar plots based on the corrected Ku- and Ka-band azimuth angle configurations (based on Figure 1(e) and (f)). We have merged Figures 10 and 11 to one single Figure 10 as shown below. There is no CPD analysis in the revised manuscript.

[Figure]

*Figure 10: Polar plot panels (a) to (f) show the relative change in averaged Ku- and Ka-band backscatter at 5° azimuth sectors, as a function of $\theta_{inc}$ , between WE1 and pre-wind conditions, acquired on 11 (WE1) and 9 November, at 2337 UTC and 0013 UTC, respectively. Panels (g) to (l) show the same between windy conditions, acquired on 15 (WE2) and 11 (WE1) November,*

*at 2338 UTC and 2337 UTC, respectively. Green arrows in (a) and (g) denote the prevailing wind direction on 11 and 15 November, respectively. The scan times also correspond to yellow circles in Figure 9 and CCTV images in Figure 5a & c. Note: The 11 November CCTV image in Figure 5c is acquired at 1736 UTC for image clarity showing blowing snow.*

Consider drawing a measurement-setup figure indicating all relevant geometric parameters like footprints, scan areas, etc. for both, Ku and Ka band.

We show the measurement-setup in figure 1(b), (e) and (f) as shown below:

[Figure]

*'Figure 1: KuKa radar geometry illustrating (a) radial distance and radar range from the pedestal foot; (b) KuKa radar azimuth scan pattern projected based on the positioner axis coordinate system (b) scan pattern of radar projected onto a level surface; (c) diameter of radar footprint, measured radially ('ra') and azimuthally ('az'); (d) area of radar footprint; (e) and (f) depicts the Ku- and Ka-band footprints of the KuKa radar, respectively. In panel (b), the region between purple lines and green lines are the respective Ku- and Ka-band*

*footprints (separately shown in panels (e) and (f), while the yellow region in the middle is the overlapping footprint area for both frequencies.'*

137: what are the scan-increments (in degree) in the azimuth direction? For incidence angle it seems to be 5°.

The azimuth scan is continuous and not discrete. We have added the word 'continuous' in section 2.1 as follows:

*"During MOSAiC, the KuKa radar scanned over a 90° continuous $\theta_{az}$ range width (between -45° and +45°) for every 5° interval in $\theta_{inc}$ between 0° and 50°.*

Section 2 or 2.2: I miss here an overview map figure (e.g. the blue sub-figure in Fig. 4 could be used) showing the location of instruments. Such a figure is very helpful for the reader to imagine where which instrument is installed and where which sampling/measurement was done.

Map on Figure 4 (with scale) clearly shows the location of the remote sensing site and its approximate distance to the snow pit locations. Therefore, we intend to use the map in Figure 4 as the overview figure and refrain from using another map since they would be seen as redundant to the maps shown in Stroeve et al. (2020).

196-202: I don't understand the azimuth sectoring method:

- why are negative and positive theta_az sectors mentioned separately? Is there any special relation between negative and positive angles?

We have thought about this comment and agree with the reviewer. With the updated geometry of the radar antennas, there is no special relation between negative and positive sector angles. Therefore, we have deleted this aspect throughout the paper and focus only on azimuth sectoring rather than distinguishing them as negative and positive sectors.

- How is the number of independent samples calculated? I would expect something like azimuth-angle-width / antenna-beamwidth * analyzed-range / range_resolution. What is "theta_az width"? Why do you devide the antenna beam-width? Why half of it? What are the range-gates? It this the range-sample spacing or the range-resolution? (note: increasing the sampling of a band-limited signal does not increas the number of *independent* samples).

We calculate the number of independent samples based on the following steps:

1. We first determine the 6-dB range swath, i.e., the distance between the 6 dB points below the peak on either side of the peak.

2. Then we divide the 6-dB range swath by the range resolution. The range resolution is 1.5 cm for Ka-band and 2.5 cm for Ku-band (full bandwidth). This is a measure of the number of independent samples in range.

3. Now we divide the azimuth width (90 and 5 degrees in our study) by the azimuth beamwidth and multiply by 2. This is a conservative estimate of the number of independent samples in azimuth.

4. The total number of independent samples would then be the number of independent samples in range multiplied by the number of independent samples in azimuth.

We have corrected the statement in the revised manuscript as follows:

*'The number of independent samples is estimated based on the following steps: a) determine the 6 dB range swath, i.e., the distance between the 6 dB points below the peak on either side of the peak; b) divide the 6 dB range by the range resolution (1.5 cm for Ka-band and 2.5 cm for Ku-band). This is a measure of the number of independent samples in range; c) divide the azimuth width (90° and 5° in our study) by the azimuth beamwidth and multiply by 2, and d) the total number of independent samples would be then the number of independent samples in range multiplied by the number of independent samples in azimuth (Doviak & Zrnić, 1984)'*

*Doviak, R. J., & Zrnić, D (1984). Doppler Radar and Weather Observations. Academic Press, 458 pp.*

Table 1: Why does the number of independent samples change with $\theta_{az}$ at Nadir? Is is $\theta_{az}$ a range of azimuth angles or does $\theta_{az}$ describe a specific angle? Or are the numbers in the table given for a 5° theta_az bin?

The numbers given in the table are for a 5° azimuth bin. We have revised the table caption as follows:

*Number of independent samples at Ka- and Ku-band frequencies at nadir and $\theta_{inc} = 50°$ at $\theta_{az} = 90°$ and along a 5° bin*

207-216: Definition of the CPD: Note, that in Leinss 2016, the CPD is derrived from S_VV * S_HH* while here it is derrived from S_HH * S_VV*, resulting in a sign change. The definition in Leinss 2016 is motivated by the desire for a positive change for increasing snow, while your definition makes more sense in terms of radar terminology. Make sure, you describe the increase/decrease of the CPD in agreement with the used definition (HH * VV* vs. VV * HH*).

We have carefully reviewed your overall comment on the utility and reliability of CPD analysis in this study and have decided to remove the entire analysis from this paper.

Figure 2: I understand that the daily wind-rose plots might be important but I miss temporally-resolved plots indicating hourly or sub-hourly resolved wind speeds. This could be plotted together with either temperature or air-pressure.

Thank you for the great suggestion. We have included the 10-min averaged air temperature (based on your next comment) and wind speed to Figure 3 as shown below.

[Figure]

Figure 2: Could you indicate in the plots, similar to Fig. 3, when the wind-events occured? This would simplify the understanding. To save space you could also plot temperature as a continuous time series (one plot, like Fig. 3a and b) with the series of daily wind-roses above or below.

[Figure]

235, Caption:

 - Surface plots -> I would say this is a 2D color plot, it does not show any 2D surface in 3D space (what I would expect for a "surface plot" in visualization-terminology. I would also avoid the word surface here, as the plot shows a time-series not a snow(or other physical)-surface.).

Thank you for the suggestion. We have changed the caption as suggested, as follows:

*Figure 3: Line plots show daily, 10-min averaged 2-m (a) air temperature, (b) air pressure, (c) wind speed and (d) relative humidity, recorded by the MET tower between 9 and 16 November. 2D color plots show DTC-derived hourly-averaged temperature gradient of (e) near-surface, snow, sea ice and ocean; and (f) sub-section of panel (e) showing the snow volume from the RSS. Yellow represents strong temperature gradients within the snowpack. Dotted red, black and white lines represent approximate locations of air/snow, snow/sea ice and sea ice/ocean interfaces. DTC temperature sensors are spaced by 2 cm, with the top 20 cm representing the*

*height above the air/snow interface. Red and orange boxes in (a) to (d) indicate WE1 and WE2 windows. Note the different temperature gradient scales for (e) and (f).'*

- "Yellow pixels represent snow volume": I know what you mean, but this statement does not make sense. Better indicate by a box or line where the snow volume is (see comment below). Yellow represents strong temperature gradients (within the snow pack).

Thank you for the suggestion. We have added dotted lines of different colours showing the estimated air/snow, snow/sea ice and sea ice/ocean interfaces. See revised Figure 3 above.

- with the top 20 cm representing the distance between the first sensor located above... and at...: I don't understand this sentence. How many sensors are above the air/snow interface? Do the top-20cm represent the height above the air/snow interface, i.e. everything above 20cm is air? How did the snow height change within the shown 8 days? Could you draw the (possibly estimated) air/snow interface into the plot?

The top 20 cm represents the height above the air/snow interface. We have corrected this in the revised caption as above.

235, Figure: Could you indicate in the plot what you consider as near-surface, snow/snow volume, sea-ice, and ocean? Could you also indicate that Fig. 3d is (I think it is) a zoom/subsection of Fig- 3c? You could indicate this by drawing the outline of the zoom into Figure 3c.

We have added lines to indicate interfaces. Also added in the caption that (f) is a zoomed in version of (e). Drawing an outline clutters the whole plot by having lines overlapping through the plot legends. The caption instead makes it clearer.

220: At which time did WE1 start?

WE1 started approximately 0745 UTC on 11 November. We have included the timing in the revised manuscript as follows: *'WE1 started ~ 0745 UTC on 11 November and lasted until ~ 0800 UTC on 12 November when winds ~12 m/s originated from the SW to SE (Figure 2).'*

252 - 253: From figure 3d there seem to be temperature gradients up to 7 or 8 K/cm during WE1, in the text I read 3 °C/cm for WE1. For WE2 fig. 3d indicates gradients of around 2-3 K/cm. Please check consistency.

We have cross checked and the temperature gradient is ~ 7°C/cm during WE1 and reducing to ~ 3°C/cm during WE2. We have revised the sentence as follows:
*'These changes clearly influenced the temperature gradients across the snowpack, with a large, vertical temperature gradient of > 7°C/cm early in WE1 decreasing to ~ 3°C/cm during WE2 (Figure 3f). Snow temperature gradients consistently exceeded 2.5°C/m, suggesting temperature gradient-driven hoar metamorphism was occurring throughout the snowpack (e.g. Colbeck, 1989).*

254: 0.25 °C/m: do you mean 2.5 °C / cm?

Corrected to 2.5°C/cm

257: uppermost: could you provide a number like e.g.: uppermost ... cm?

Uppermost 2 cm snow layers. Corrected in the revised manuscript.

274: I doubt that breakup of snow particles decreases the SSA. No matter how SSA is defined, as surface area per volume or kg, breaking up crystals would increase the SSA because the grain size get's smaller by the breaking events. King 2020 describes the SSA decrease in wind slaps rather as a "product of mechanical wind rounding and subsequent sintering".

We agree with the reviewer and have revised the sentence as follows: *'A SSA decrease indicates the reduction in surface area, caused by rounding of snow grains, followed by sintering during wind transport (King et al., 2020).*

305: "superimposed on the TLS data in yellow ... and orange where the two overlap": I understand the yellow-to-blue colors in Figure 6 so that black indicates no TLS data (count=0). Why are there then a yellow or orange colored radar footprints or TLS data that are located on black pixels? Or does yellow and red indicate two different radar observations? Please clarify in the caption what is yellow, orange and red. As mentioned earlier, I did not understand the difference in radar observations in the positive and negative azimuth angles theta_az.

Thank you and we agree that the choice of colours was not good. We have changed Figure 6 and the caption, as follows.

[Figure]

*'Figure 6: TLS data (plan view) from 1, 8 and 15 November, from -90° to + 90°, where the angle indicates the azimuth of the radar positioner, and radial horizontal distance measured from the centre of the radar pedestal. The top panels show the topography as measured downwards (increasing negative) from the middle of the radar antenna arms. Black indicates no data recordings in that bin. Projections of the centres of the radar footprints are shown for 0° and 50° radar incidence angles between -65° to + 65° azimuth range, superimposed on the TLS data in magenta and green for radar observations, respectively, and buff where the two overlap, as per Figure 1. The bottom panels indicate the number of TLS data points within each bin. Surface depressions resulting in 0 counts in the TLS data are due to obscuration by adjacent high areas due to snow/sea ice topography and human-made objects, as viewed from the TLS's oblique viewpoint some distance away.'*

Figure 6 and Figure 7: The TLS data and the radar-Nadir data indicate a possibly considerable slope within the observation area. Looking at both figures, I can estimates slopes of 2-5°. Would it be possible to make any statement about changes of the local slope in the observation area? As observed later, in Figure 9, the incidence angle has a very significant effect on the radar waveform, hence the local incidence angle must also have an significant effect. I can't tell of 2-5° are already significant, but I belive they are.

Thank you for the observation. Yes, the local incidence angle is particularly important at nadir or near-nadir backscatter. Sloped surfaces of 2-5° will significantly affect the total backscatter magnitude. However, since surface scattering is the dominant scattering mechanism at nadir and steep incidence angles, slightly sloped surfaces that we observe in this study likely do not affect the relative distribution of scattering between the air/snow and the snow/ice interface scattering.

We have added this observation in section 3.2 as follows:

*The TLS and radar waveforms also indicate a ~ 2-5° slope in the radar scan area especially at nadir (See Figures 6 and 7). Sloped surfaces of 2-5° will significantly affect the total backscatter amplitude. However, since surface scattering is the dominant scattering mechanism at nadir, slightly sloped surfaces observed from the radar scan area likely do not affect the relative distribution of scattering between the air/snow and the snow/ice interfaces.*

Figure 7: I find it quite confusing that Figure 7 shows TLS profiles from different dates in every figure. Why not showing the TLS profile from 08 Nov (and possibly 01 Nov) for the radar data from Nov 09 , 11, and possibly 11 and the TLS data from Nov 15 ontop of the radar data from Nov 15? If you perfer to keep all three TLS profiles, then please mention why different TLS profiles are shown ontop of the radar data for each date.

TLS data were gathered weekly so the 8 and 15 November are the only available TLS datasets within the study period. The 8 November has considerable missing data due to the positioning of the TLS scanner and we therefore use the 1 November as a similar dataset (pre wind event) with more data points, as noted in the text. We also show the 15 November on the same plots

so that the before and after elevations of the air/snow interface are present, to aid comparison with the KuKa radar data for all dates.

378, caption 8:

 - "red, yellow, black": I do not see a yellow line in the figure. Consider refering only to the dashed red and black lines.

We have rephrased to clarify.

 - But keep the sketched yellow arrows in the figure (very interesting!) and the sentence refering to them.

Thank you. These are kept.

 - Mention the meaning of the vertical gray lines in the caption, I guess they constrain WE1. You could also add a label "WE1" to the figure.

We have added this label and in the caption it is also mentioned

 - "Time series of the interfaces NRCS values are snown below the echograms" -> "NRCS time series of the two interfaces (red, black) are snown below the echograms for HH and HV (dashed/dotted)". Also add a legend to the timeseries indicating the colors and line-styles.

We already have a legend showing this and adding legends for colors and linestyles could clutter the already busy plot. Therefore, we wish to retain the presently used legend in the plot.

[Figure]

- Figure 8 has many sub-panels. Consider labeling them with (a, b, c...). Is figure 8, bottom right mentioned in the text? If not, it seems not to be too important and could be removed.

We have labelled the panels. We would prefer not to remove any of the panels as these form the complete set of co- and cross-pol data for the reader's reference.

 - Figure 8 (and other figures): Consider labeling the time-axis according to the shown tick labels, e.g. "Date (YYYY-MM-DD)" or "Date / time (MM-DD HH:mm)"

We have adjusted the labels to what we hope is a clearer format.

356: "It is interesting ...can still be seen ... 10 November, .." add a reference to "yellow arrows, Figure 8".

Added.

359-360: "During WE2, ... the air/snow interface moved upwards... (bottom right of Figure ... and 8)": I do not see any effect of WE2 in Figure 8. The x-axis of Fig. 8 ends likely at end of the day 2019-11-15. Could you indicate, similar to WE1 by gray vertical bars, where WE2 is located in Figure 8?

We have added this.

393 and 394: Both sentences: Add reference to the corresponding panels in Figure 8.

Included.

415-416: "... at all theta_inc. VV and HH backscatter primarily originates as surface scattering at the air/snow interface" I think this might be an inadmissible generalization. The fact, that Nadir observations indicate the strongest return at the air/snow interface cannot be generalized to all incidence angles. On the contrary, the backscatter at the air/snow interface might even be reduced for non-zero incidence angles due to specular reflection away from the radar.

We agree with the reviewer and have removed this generalisation, as follows:
*During pre-wind conditions, both Ka- and Ku-band backscatter are relatively stable (Figure 9a & b). At nadir, VV and HH returns primarily originate from the air/snow interface. With higher values of  , air/snow interface scattering decreases due to the specular component of the backscattering not returning to the radar detector. The signal is therefore increasingly dominated by snow volume scattering and incoherent surface scattering at the snow/sea ice interface. HV backscatter originates primarily from the snow/sea ice interface (top panels in Figure 7).*

420-429: This seems more an interpretation of the Nadir observations in Figure 7 and 8 rather and seems less relevant or even misleading for the interpretation of off-Nadir observations. Consider moving to the interpretation of figure 7 and 8 and refer only briefly to this interpretation when discussing the non-nadir angles in Figure 9.

We disagree with the reviewer on this comment. Our objective in this section is to link the changes in azimuthally averaged NRCS at nadir, 15, 35 and 50 incidence angles to what we see from the waveform analysis. Moving the nadir section to section 3.2 will isolate nadir NRCS observations and focus ONLY on the non-nadir incidence angles.

436-439: I think observation of different layers at non-zero incidence angles is difficult due to the slant imaging geometry. In the slant geometry each slant-range bin samples the backscatter from all targets located at the same range (bin). This includes contributions from the air/snow surface, the volume and the snow/ice interface. As for each slant-range distance a slant cross section through the scattering volume is measured, the measured profile is rather a representation of the beam-pattern weighted by the incidence-angle dependent backscatter intensity from the different surfaces illuminated by the beam.

Similar to Figure 9 and 10 in [Leinss et al. EUSAR 2014] I interpret the strong near-range return in Figure 9c (at theta_inc=15°), possibly also 9d (30 deg) as a Nadir-return from the air/snow interface (consider the beam-width of 12-17°). However, the increasing slant-range distance to the strongest return is a good indicator that the snow above the ice delays the radar signal.

439/440: "The waveform analysis shows": Please be more specific. Similar to the observations with the SnowScat instrument, where dry arctic snow up to 1m depth appeared almost transparent at incidence angles between 30 and 60° and X to Ku-band frequencies [https://doi.org/10.1109/JSTARS.2015.2432031], I interpret the increasing range to the strongest return caused by 1) increased delay to to increased accumulation and 2) caused by the snow/ice interface. See also Figure 9 and 10 in [Leinss et al. EUSAR 2014] where the snow/soil interface is detected. This interface appears further away from the sensor (due to increasing delay) with increasing snow depth.

Thank you for the above 3 comments. We agree with the reviewer as to how increased snow accumulation can delay the propagation speed leading to increased ranges at shallow angles. We have corrected our interpretation as follows (answering all three comments above into one):

*The effect is less at 35° due to the snow volume scattering becoming more dominant compared to surface/interface scattering at the slanting cross section at more oblique angles. The waveform analysis shows that the relative contribution of the snow/sea ice interface, snow volume scattering and increased radar propagation delay due to increased snow accumulation becomes more important at shallow angles (Leinss et al., 2014) and the air/snow interface becomes relatively less prominent due to lower surface roughness after WE1.*

Leinss, S., Lemmetyinen, J., Wiesmann, A., & Hajnsek, I. (2014, June). Snow Structure Evolution Measured by Ground Based Polarimetric Phase Differences. In *EUSAR 2014; 10th European Conference on Synthetic Aperture Radar* (pp. 1-4). VDE.

Reference: Leinss, Silvan and Lemmetyinen, Juha and Wiesmann, Andreas and Hajnsek, I.. (2014). Snow Structure Evolution measured by Ground Based Polarimetric Phase Differences.

in Proceedings of European Conference on Synthetic Aperture Radar (EUSAR)]
https://www.researchgate.net/publication/312171448_Snow_Structure_Evolution_measured_
by_Ground_Based_Polarimetric_Phase_Differences

We have added this reference in the revised manuscript.

Figure 10, 11, 12: These figures take up a lot of space but seem to reveal little information. Consider removing them (and the associated discussion) if you agree that they are not crucial for the main points of the manuscript.

We do agree with the irrelevance of co-pol phase difference analysis (Figure 12) in this paper using the polar plots thus we have removed the entire analysis from the paper.

However, we have retained the backscatter analysis using azimuth sectoring as the polar plots (we have merged Figures 10 and 11 into one single Figure 10) clearly show spatial variability in backscatter within the footprint, as a function of incidence angle and polarization, in response to wind-driven snow dynamics and thermodynamic-induced metamorphic changes to snow layering. This analysis shows the importance of accounting of spatiotemporal variability in snow properties within a satellite footprint that can induce associated changes in dominant scattering horizon and backscatter. In the revised version, we have revised this analysis through improved representation of KuKa radar geometry.

451: "Next, we show..." Sentence should be used as introductory sentence for section 3.3.2 and not at the end of section 3.3.1.

Sentence moved to the beginning of section 3.3.2 as suggested.

482: "stable snow metamorphism": metamorphism is a dynamic process. Do you mean "stable snow conditions with little metamorphism" or "continuous metamorphism"?. Variations of the CPD indicate metamorphism, however a single observation of the CPD does not provide any information about the dynamics of the snow pack. Nevertheless, it can give an idea about the history of methamorphism of the snow pack.

This sentence belongs to the CPD analysis. The whole analysis is removed in the revised version.

499: What is "phase reversal"? Do you mean "phase wrapping"?

CPD analysis removed from the revised version

480-502: I have some doubts on the results and interpretation of the CPD.

CPD analysis removed from the revised version

- First of all, the backscatter results in the previous sections are most convincing when averaged over azimuth (but not over incidence angle!) and plotted with temporal resolution (like Figure 8 and 9). Why not showing such plots for the CPD first (at different incidence angles, because

the CPD is strongly incidence angle dependent)? If temperature-gradient-metamorphism induces variations of the structural anisotropy then these variations should be well visible during the extreme temperature gradients up to 800 K/m shown in Figure 3.

CPD analysis removed from the revised version

- Could the authors ensure that they use the same definition of the CPD as in Leinss 2016? See comment above: Currently, there might be a sign error. The sign of the CPD might also dependent on the data processing and chosen side-bands in the electronics of the instrument.

CPD analysis removed from the revised version

- Did the authors ensure that the CPD of the instrument was well calibrated? Snow-free data and open water should have zero CPD.

CPD analysis removed from the revised version

- An analysis of CPD time series, averaged over the whole area might also indicate calibration issues.

CPD analysis removed from the revised version

- The instrument seems to allow processing of the data at user-defined bandwidth and central frequencies. Phase wraps can be easily detected by using two slightly different frequencies. See approach in [https://doi.org/10.1109/JSTARS.2015.2432031].

CPD analysis removed from the revised version

- At Ka-band, there seem to be large phase differences at Nadir. It could be speculated that this could be caused by an wind-induced anisotropy in the x-y-plane (horizontal) rather than processes that act in the vertical direction (settling, metamorphism). However, a non-zero CPD at Nadir might also indicate a not well calibrated instrument. Please check.

CPD analysis removed from the revised version

504: "The dominant radar scattering surface": add: "at nadir and for both Ku- and Ka-band"

Added as suggested.

513-514: "provide contextual information for reliable interpretation": this is a very vague statement. Try to draw more specific conclusions. One point I see is: The fact that the dominant radar return for incidence angles > 15 degree, possibly even smaller, is not the air/snow interface anymore has an important impact for Altimeter-observations on snow-covered slopes. For such slopes, the surface height is likely to be underestimated. If you agree, please mention it in the discussion.

Thank you for the suggestion. We have revised the sentence as follows:

*'At satellite scales, this can likely overestimate sea ice freeboard when assuming that the snow/sea ice interface is the dominant scattering surface, and therefore, warrants careful interpretation of waveforms and backscatter at nadir. At non-nadir incidence angles, the relative scattering contribution of the snow/sea ice interface compared to the air/snow interface increases, and the air/snow interface gradually becomes invisible (Figure 9), as a result of the snow surface smoothening.'*

517: "which indicates that snow density and surface roughness contrasts (Figure 4) existing prior to ..." -> "which indicates that snow layers existing prior to ...": I don't understand why surface roughness is mentioned here. The manuscript shows no data on surface roughness; Especially not in Figure 4. What does the word contrasts refer to? I don't understand. Consider revising this sentence.

It is unfortunate that surface roughness data are not available for this study, and as noted above we have added context for this in both the discussion and conclusion: based on previous studies, roughness is an important parameter to quantify and that this data should be collected in future studies.

523: "The relatively small backscatter indicates ..." I think, the relatively small backscatter is a consequence from the specular reflections away from the sensor at non-nadir angles. It's rather the increasing delay (slant-range) to the observed echo that indicates that most scattering is associated with the snow/ice interface. See comment above (436-439).

We have revised the statement as follows: *'The relatively small backscatter observed from the snowpack at $\theta_{inc} = 15°$ and 35° (Figure 9c & d) indicates dominant scattering away from the radar. Additionally, at these angles, most of the backscatter is associated with the snow/sea ice interface, and that deeper snow is causing an increasing slant-range delay.'*

524: what are "shallow theta_inc"? Incidence angles close to zero or close to 90°?

We meant non-nadir incidence angles and have changed accordingly in the revised manuscript.

524/525: What does "change" mean here? What changes under which conditions?

We refer to this change in snow volume scattering to changes in microstructure in response to wind-driven thermodynamics and its impact on the snow pack. We have revised the statement to:

*'This absence of volume scattering change (due to wind-driven snow microstructural changes) at non-nadir $\theta_{inc}$, in combination with the observed nadir sensitivity, suggests that surface scattering is the dominant scattering mechanism at nadir.'*

535: "a two-scale function of the microscale surface roughness" what do you mean? What is a two-scale function?

The 'two-scale' function refers to the RMS and correlation length components of the surface roughness. To avoid confusion, we have removed this phrase.

569/570: I doubt that "strong contributions from snow grain volume scattering at C-band" exists.

If the storm has a thermodynamic effect on snow-covered first-year sea ice, then C-band volume scattering contributions from brine inclusions within the snowpack cannot be ruled out. We have noticed this effect during calm conditions over saline snow on first-year sea ice from Ku, X and C-band scatterometry (e.g. Nandan et al., 2016)

Nandan, V., Geldsetzer, T., Islam, T., Yackel, J. J., Gill, J. P., Fuller, M. C., ... & Duguay, C. (2016). Ku-, X-and C-band measured and modeled microwave backscatter from a highly saline snow cover on first-year sea ice. *Remote Sensing of Environment*, *187*, 62-75.

584: why would the anisotropy induce "scale-dependent" properties? I think scale-dependent should be removed here.

CPD analysis removed from the revised version

585: Why would the anisotropy alter surface and interface roughness? I would rather agree that snow metamorphism can alter interface roughness.

CPD analysis removed from the revised version

586-596: Revise according to the outcomes resulting from adressing comment 480-502.

CPD analysis removed from the revised version

599: "the first-ever recording of" a matter of taste if you need that or not. As publications should naturally contain new and original data all main results should be first-ever anyway. I would remove this statement.

We have removed this statement.

603-604: Revise sentence to make it accurate.

- As observed in Figure 9, the air/snow interface can be hardly detectable with non-zero incidence angles. I even presume that the air/snow signal at 15° is a Nadir-return of the 12-17° beam-width (6 dB two-way beamwidth according to Strove 2020). As the beam-width does not have sharp edges it is certain that a Nadir-return can be observed.

- "buried air/snow interface remains detectable" well, yes, this is shown in the nadir-looking data in Figure 8. However, this figure also shows that the intensity of the buried air/snow interface seems to be 10-20 dB below the intensity of the air/snow interface.

Yes, we agree with this and have used the term 'detectable' to indicate that, although at far lower power, the KuKa radar is still capable of detecting these features.

613: "strong spatial variability in backscatter": The figures 10 and 11 indicate temporal changes of less than 3 dB for WE1, and around 5 dB for WE2 with an decrease in Ka and an increase in Ku band. However as change and no absolute backscatter is shown one cannot speak about a strong spatial variability, rather than about a temporal variability of 3-5 dB at least for the second wind event.

We somewhat disagree with the reviewer's comment here. In our azimuth sectoring analysis, we show the relative change in backscatter as a difference between two instances of a) difference between WE1 and calm conditions (old Figure 10) and, b) difference between WE2 and WE1 (old Figure 11). In these analyses, we show the differences in backscatter at all incidence angles and polarizations, across the 5 degree azimuthal sector bins. Our point is that when we average backscatter across the entire azimuth, the variability in geophysical change is also averaged. Or in other words, the variability in the dominant scattering surface is also masked. We have modified the sentence as follows in the revised manuscript:

*'Compared to pre-wind conditions, nadir backscatter across the full radar azimuth increased by up to 8 dB (Ka-band) and by up to 5 dB (Ku-band) during the wind events. This was caused by the formation of snow bedforms within the radar footprint, which increased the snow surface roughness and/or density. Azimuth sectoring at 5° bins reveals the spatial variability in backscatter across the radar footprint, in response to formation of snow bedforms caused by increasing wind speeds and changing wind direction.'*

614-617: Revise according to the outcomes resulting from adressing comment 480-502.

We have removed CPD analysis from this manuscript.

Technical comment:

36/37: "changes in ...backscatter coincident with ... are observed": The sentence is correct, however, I had to read it a few times to make sure that really "backscatter coincident with..." is meant and not "backscatter coefficient". You might want to move the verb before coincident: "changes in ...backscatter (coefficients) are observed, coincident with ... "

Corrected

39: detect -> detected

Changed as suggested

47: "Our results reveal the imprtance of wind, through its geophysical impact on Ka- and Ku-band (...) and has implications..." I'd suggest: "Our results reveal the impact of wind on Ka- and Ku-band (...) which has implications..."

Changed as suggested

68: I'd suggest: -> will result in the formation of heterogenities on different scales, from cm-scale ripple marks to snow bedforms ...

Corrected as suggested

131: from "nadir to theta_inc = 50°" maybe, from "theta_inc = 0 - 50°"

Corrected as suggested

142: "denote d" -> "denoted \textit{d}" (italic d)

corrected

175 "across the theta_az range" -> "across theta_az" (here and other places: Even though linguistically correct, I would avoid using the word "range" together with "azimuth" to describe a span/interval/sector to avoid confusion with slant-range of the radar)

Corrected throughout the manuscript

234: "are spaced every 2 cm" -> "are placed every 2 cm" or more accurate: "are spaced by 2 cm".

corrected

256: increase and decrease in density and SSA -> increase in density and decrease in SSA

corrected

525: "dominant changing scattering mechanism" add: "at nadir".

Added.

527: "changed by more " -> "increased by more"

Corrected.